# Convergence dynamics of Agent-to-Agent Interactions with Misaligned objectives

## Abstract

We develop and analyze a theoretical framework for agent-to-agent interactions in a simplified in-context linear regression setting. In our model, each agent is instantiated as a single-layer transformer with linear self-attention (LSA) trained to implement gradient-descent-like updates on a quadratic regression objective from in-context examples. We then study the coupled dynamics when two such LSA agents alternately update from each other's outputs under potentially misaligned fixed objectives. Within this framework, we characterize the generation dynamics and show that misalignment leads to a biased equilibrium where neither agent reaches its target, with residual errors predictable from the objective gap and the prompt-induced geometry. We further contrast this fixed objective regime with an adaptive multi-agent setting, wherein a helper agent updates a turn-based objective to implement a Newton-like step for the main agent, eliminating the plateau and accelerating its convergence. Experiments with trained LSA agents, as well as black-box GPT-5-mini runs on in-context linear regression tasks, are consistent with our theoretical predictions within this simplified setting. We view our framework as a mechanistic framework that links prompt geometry and objective misalignment to stability, bias, and robustness, and as a stepping stone toward analyzing more realistic multi-agent LLM systems.

## 1 Introduction

Large language models (LLMs) increasingly act as *agents* that exchange messages, propose edits, and iteratively refine solutions in multi-step workflows (Mohammadi et al., 2025; Niu et al., 2025; Zhang et al., 2025). While this trend has spurred a surge of multi-agent designs, from debate and role-structured discussions to autonomous tool-using collectives (Wu et al., 2024; Du et al., 2023; Liang et al., 2024; Chen et al., 2023), their behavior remains difficult to predict, especially when agent goals are only partially aligned (Erisken et al., 2025; Altmann et al., 2024; Cemri et al., 2025; Kong et al., 2025). Recent empirical findings further caution that, under common prompting and coordination schemes, multi-agent setups may not consistently outperform strong single-agent baselines and can be brittle and unreliable participants (Wang et al., 2024; Huang et al., 2025a; Wynn et al., 2025; Lee & Tiwari, 2024). These observations motivate a principled, mechanistic account of how interacting LLM agents update their internal states *because of* each other.

Our analysis builds on an emerging theoretical view of LLM inference as *in-context optimization*. A growing body of work shows that sufficiently trained transformers can implement algorithmic updates, including gradient descent for linear regression tasks, using only the information provided in the prompt (Akyürek et al., 2023; Garg et al., 2022; von Oswald et al., 2023; Ahn et al., 2023; Dai et al., 2022). Most relevant to us, Huang et al. (2025b) prove that a single-layer transformer with linear self-attention (LSA) can carry out *multiple* gradient-descent-like steps in context when trained to predict the next iterate on quadratic objectives. We adopt this insight as a modeling primitive: specifically, we assume that once appropriately trained, each agent performs a stable, approximately linear *gradient* update towards its own objective, from the incoming context (representing the previous iterate). In the rest of the paper, "agent" refers specifically to such an LSA-based in-context optimizer operating on a linear regression objective. We use this analytically tractable model as a proxy for LLM-based agents

Building on this "transformers-as-optimizers" perspective, we theoretically investigate *agent-to-agent* interactions as an *alternating optimization* process between two LSA agents with potentially misaligned objectives. Concretely, at each turn an agent consumes the other's latest iterate and applies a gradient update towards its own objective. In the resulting *fixed-objective* multi-agent regime, the coupled dynamics converge to biased fixed points whose residuals are jointly governed by (i) **objective misalignment** (the discrepancy between objectives) and (ii) **prompt geometry anisotropy** (spectral structure of agent-specific covariances that shape update directions); anisotropy induces directional filtering, amplifying each agent's error along directions dominated by the *other* agent's geometry. We also contrast this regime with an *adaptive* multi-agent regime in which a helper agent updates a turn-based objective and can implement Newton-like steps for the main agent, turning the same interaction formalism into a mechanism for cooperative acceleration rather than mutual degradation.

We then characterize the conditions under which the agent-to-agent dynamics admit *asymmetric convergence*: where one agent can attain its objective exactly while the other is left with a persistent bias. These conditions translate into constructive mechanisms for *adversarial prompt design* that cancel an opponent's corrective directions while preserving the attacker's progress. This connects predictive modeling to concrete security concerns for multi-agent LLM systems (He et al., 2025; Struppek et al., 2024; Xi et al., 2023; Wang et al., 2023).

We validate the theory with experiments using trained LSA agents in the sense of Huang et al. (2025b). We also provide experimental validations with GPT-5-mini for our adversarial prompt design approach. Importantly, when objectives align, the shared iterate converges cooperatively to the common objective. Under misalignment, both agents plateau at analytically predicted, generally *unequal* residuals that grow with the inter-objective angle. Under adversarial designs derived from our kernel criteria, we observe reliable asymmetric outcomes: the attacker converges to its objective while the victim remains biased.

Our contributions are summarized as follows: **(i)** We formalize agent-to-agent interactions as alternating, in-context gradient updates between two transformer agents (Section 2). **(ii)** We obtain closed-form expressions for each agent's limiting error that depends on global objective misalignment and prompt anisotropy. We also include spectral analysis of the error and derive error bounds with respect to the angle between the global objectives. We also extend the analysis by introducing local objectives and further demonstrate how a collaborative agent can accelerate convergence of the main agent beyond what the main agent can achieve by itself. (Section 3). **(iii)** We establish kernel conditions for *asymmetric convergence* and give a constructive adversarial geometry that enforces it leading to a white-box attack procedure (Section 4). **(iv)** We corroborate these theoretical results with trained LSA agents as well as GPT-5-mini experiments, highlighting when and how multi-agent interactions can be helpful, when they result in agent compromises, and when they can be steered to harmful outcomes. While the experiments are provided throughout the paper, experimental details are given in Section 5.

## 2 AGENT-TO-AGENT FORMALISM

In this section we develop a formal model of *agent-to-agent* interactions grounded in the emerging view of LLM inference as *in-context optimization*. Rather than analyzing prompting procedures directly, we consider that each agent realizes a *gradient* update on its own objective from the received context. This assumption is supported by theory and experiments showing that trained transformers can implement algorithmic updates, including multi-step gradient descent for quadratic objectives, purely in context; in particular, Huang et al. (2025b) establish such behavior for single-layer LSA.

We first recall the single-agent setting in which an LSA model, given a dataset packaged as tokens, emits successive iterates that track gradient descent on a least square regression problem. We then lift this formalism to propose a theoretical backbone to agent-to-agent interactions. In such case, which each agent has its own set of weights, and a specific prompt dependent on its linear regression objective. The agents interact by alternating turns; each consuming the other's latest iterate and applying its own in-context update toward its objective. The result is a coupled, turn-by-turn dynamical system amenable to fixed-point and spectral analysis. This agent-to-agent framing allows us to quantify how *objective misalignment* and *prompt geometry* jointly determine convergence, plateaus, and potential asymmetries.

## 2.1 IN CONTEXT OPTIMIZATION

Chain-of-Thought (CoT) prompting (Wei et al., 2022) enables large language models to break down complex reasoning into intermediate steps, significantly improving performance on mathematical and logical tasks. Recent theoretical work has revealed the optimization foundations underlying this process. Huang et al. (2025b) provide a theoretical analysis of how transformers can learn to implement iterative optimization through CoT prompting. They consider a linear regression task within the in-context learning (ICL) framework and demonstrate that a suitably trained transformer can perform multiple steps of gradient descent on the mean squared error objective.

The data consist of $n$ example input-output pairs from a linear model,

$$w^\star \sim \mathcal{N}(0, I_d), \qquad x_i \sim \mathcal{N}(0, I_d), \qquad y_i = x_i^\top w^\star \quad \text{for } i = 1, \ldots, n.$$

The learner is given these examples in context and must estimate the underlying weight vector $w^\star$ (without further gradient updates to its own weights). The key insight is that a transformer can use CoT to iteratively refine an internal estimate of $w^\star$ over $k$ autoregressive steps.

The LLM is modeled as a single-layer LSA transformer with residual connections. The input to the LSA is as follows:

$$Z = \begin{bmatrix} x_1 & \cdots & x_n & 0 \\ y_1 & \cdots & y_n & 0 \\ 0 & \cdots & 0 & w_0 \\ 0 & \cdots & 0 & 1 \end{bmatrix} := \begin{bmatrix} X & 0 \\ y & 0 \\ 0_{d \times n} & w_0 \\ 0_{1 \times n} & 1 \end{bmatrix} \in \mathbb{R}^{d_e \times (n+1)},$$

where $X = [x_1, \ldots, x_n]^T \in \mathbb{R}^{n \times d}$ is the data matrix, $w_0 = 0_d$ is the initialization of the objective weight, and $d_e = 2d + 2$. Note that the token matrix $Z$ encodes input data $(x_i, y_i)$ and also includes dimensions to autoregressively represent the current weight estimate.

The LSA mapping is defined as:

$$f_{\text{LSA}}(Z; V, A) = Z + VZ \cdot \frac{Z^\top A Z}{n},$$

where $V, A \in \mathbb{R}^{d_e \times d_e}$ are learned weight matrices. The model's prediction is the embedding of the final token

$$w = f_{\text{LSA}}(Z)[:, -1].$$

With appropriate training, the LSA transformer learns to output a sequence of weight estimates $\{w_0, w_1, \ldots, w_k\}$ where each CoT step approximates a gradient descent update,

$$w_{t+1} \approx w_t - \eta \frac{1}{n} X^\top (X w_t - y), \tag{1}$$

with $\eta > 0$ the learning rate. In other words, at each CoT step, the LSA transformer performs a gradient descent step on the least square loss $\frac{1}{2}\|Xw - y\|^2$ with respect to its previous weight estimate.

It is important to note that the affine updates under study in this paper do not rely on architectural linearity of the agents but on the linear regression objective. For quadratic losses, the gradient-descent rule is inherently affine, and any sufficiently expressive model trained to perform such in-context optimization (including full transformers) will implement an affine update in the representation space. Thus, the linearity here reflects the structure of the task-level gradient dynamics, not a simplification of model architecture.

## 2.2 AGENT-TO-AGENT FORMULATION

We now extend this framework to agent–to–agent interactions under an alternating turn-taking protocol. In this setting, two agents engage in a dialogue where, at each turn, an agent receives as input the prompt and accumulated conversation history, and subsequently generates an output response.

Consider two agents, $W$ and $U$, that alternate turns: each agent receives the other's output and performs one step toward its own objective. Following the aforementioned linear regression formalism,

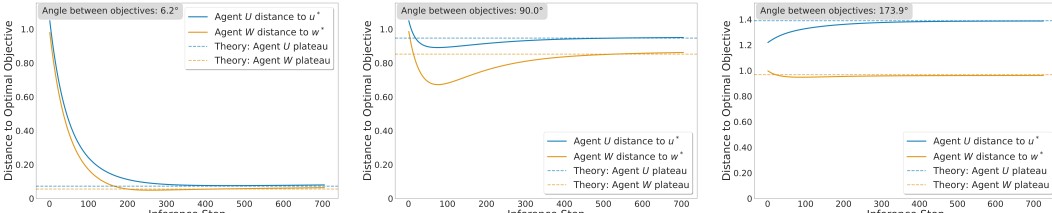

Figure 1: **Plateau error vs objective alignment:** (*left*) With aligned objectives, both agents converge cooperatively to the shared objective. Note that because of the $\sim 6°$ angle between objective, the agents do not converge to 0-error. (*middle*) With orthogonal objectives ($\sim 90°$), convergence occurs toward a solution that does not advantage either agent. (*right*) With opposite ($\sim 174°$) objectives, the dynamic is similar to the orthogonal objective case. Note that ($i$) whether agent $U$ or agent $W$ converges to a better error is induced by the prompt geometry, and ($ii$) in all cases here, neither agent converges to a 0-error solution. These two key points are central to the characterization we provide in Section 3.

we consider the following data structure at turn $t$,

$$
Z_W = \begin{bmatrix} X_W & 0 \\ y_W & 0 \\ 0_{d \times n} & u_0, w_1, u_1, \ldots, u_{t-1} \\ 0_{1 \times n} & 1 \end{bmatrix}, \quad Z_U = \begin{bmatrix} X_U & 0 \\ y_U & 0 \\ 0_{d \times n} & u_0, w_1, u_1, \ldots, u_{t-1}, w_t \\ 0_{1 \times n} & 1 \end{bmatrix},
$$

In this construction, agent $W$ utilizes $(X_W, y_W)$ together with the conversation history $[u_0, w_1, u_1, \ldots, w_{t-1}, u_{t-1}]$ to produce the update $w_t$. Now, agent $U$ employs $(X_U, y_U)$ along with the extended history $[u_0, w_1, u_1, \ldots, w_{t-1}, u_{t-1}, w_t]$ to generate $u_t$. Note that we default the initialization to $u_0 = 0_d$ and consider that agent $W$ speaks first.

Note that, each agent may have different objectives. In our theory that takes the form of having misaligned regression objectives $w^\star \neq u^\star$. Building on Huang et al. (2025b), there exists a parametrization of the LSA under which each mapping applied to the input data approximates a gradient descent update. Such a parametrization arises from training the LSA toward the gradient-descent update. All LSA experiments in this paper are inference-only and use LSA agents that were pretrained (in a single-agent setting) to generalize the gradient–prediction task described in Section 2.1.

Consequently, each agent also admits the gradient-descent update defined in Eq. 1. The resulting alternating dynamics between the two agents that will be central to this paper are given by

$$
w_{t+1} = u_t - \eta S_W (u_t - w^\star) \tag{2}
$$

$$
u_{t+1} = w_{t+1} - \eta S_U (w_{t+1} - u^\star), \tag{3}
$$

where $S_W = \frac{1}{n} X_W^\top X_W$ and $S_U = \frac{1}{n} X_U^\top X_U$ the covariance matrices of the data. When the agents pursue aligned objectives, i.e., $w^\star = u^\star$, these alternating updates collapse to the single agent formalism as defined in Huang et al. (2025b). *In contrast, when objectives are misaligned ($w^\star \neq u^\star$), the agent-to-agent dynamics may give rise to different behaviors, including mutual convergence, asymmetric convergence (where one agent achieves its objective while persistently biasing the other), or adversarial interactions in which one agent systematically manipulates the trajectory of the conversation.* The remainder of the paper is devoted to analyzing these interactions at inference given trained models.

## 3 AGENT-TO-AGENT DYNAMICS

In this section we study the alternating agent-to–agent update dynamics in our in-context linear regression model. We first analyze the *fixed-objective* multi-agent regime, where each agent's target $(w^\star, u^\star)$ is held constant, and derive *explicit* expressions for their asymptotic errors, which explain the unequal convergence plateaus observed when two misaligned agents interact (see Figure 1) and

yield angle-based bounds (Figure 2). We then turn to an *adaptive* multi-agent regime in which a helper agent updates a turn-based objective for the main agent and can implement Newton-like accelerated steps (Figure 3). Throughout, we assume convergence to a fixed point, which imposes a standard condition on the gradient-descent stepsize.

The following proposition characterize asymptotic errors of each agent from their respective objectives as a result of turn-base agent-to-agent interaction at inference with a trained LSA model.

**Proposition 1.** *Let $S := S_W + S_U$ be invertible and let $\Delta = u^\star - w^\star$, then as $\eta \to 0$,*

$$\|u_\infty - u^\star\|_2^2 = \Delta^\top (S_W S^{-2} S_W) \Delta + O(\eta), \qquad \|w_\infty - w^\star\|_2^2 = \Delta^\top (S_U S^{-2} S_U) \Delta + O(\eta). \quad (4)$$

*(Proof in Appendix 9.2)*

Assuming $S$ is invertible means that there are no blind directions where the misalignment $\Delta = u^\star - w^\star$ can hide from both agents. In practice, one ensures invertibility by using sufficiently diverse, non-collinear in-context examples across the two prompts.

This proposition shows that, after sufficiently many turns, each agent's residual error is governed by two key factors: (i) the discrepancy between the agents' objectives, and (ii) the structure of their respective prompts. In the linear regression setting, that is, the covariance structure of the data. Note that the squared asymptotic errors capture the smoothness of the objective difference, i.e., $\Delta$, along the spectrum of $S_W$ (resp. $S_U$) normalized by $S$. Therefore, the anisotropy of $(S_W, S_U)$ can potentially make these plateaus unequal, leading to agent-to-agent dependent convergences.

In an LLM, an analogous notion of prompt geometry can be defined directly in representation space. Given a prompt $P = (t_1, \dots, t_n)$ and its embedding representations $h_1, \dots, h_n \in \mathbb{R}^d$, one can form a second-moment matrix $S(P) = \frac{1}{n} \sum_{i=1}^n h_i h_i^\top$, whose dominant directions and anisotropy reflect which linguistic structures (semantic fields, styles, task types) are repeatedly instantiated by the prompt. For example, a prompt dominated by arithmetic expressions, by code snippets, or by legalistic text will emphasize different subspaces in representation space. In our LSA model, the matrices $S_W$ and $S_U$ play this role for the feature vectors appearing in each agent's prompt. Recent empirical taxonomies of multi-agent failures, such as MAST (Cemri et al., 2025), identify *system design issues*, including flawed role specifications and ambiguous prompts, as a primary source of breakdown; in our framework, these design choices manifest as misaligned effective objectives $(w^\star, u^\star)$ and ill-designed geometries $(S_W, S_U)$, and Proposition 1 shows that such misalignment inevitably induces biased plateaus even when each agent is individually competent.

In Figure 1, we observe at inference the empirical error of each agent towards their objective as well as the computed theoretical convergence plateau obtained from Proposition 1. Importantly, the asymptotic error can be computed before any agent-to-agent interaction given knowledge of the prompts and the objectives.

The quadratic forms in Proposition 1 highlight that the limiting plateaus are not determined solely by the objective misalignment $\Delta$, but also by the anisotropy of the agents' prompt geometries $(S_W, S_U)$. In the isotropic case, where $S_W$ and $S_U$ are multiples of the identity, the weights $S_W S^{-2} S_W$ and $S_U S^{-2} S_U$ collapse to scalars, and both agents experience identical plateau errors proportional to $\|\Delta\|_2^2$. By contrast, when the spectra of $S_W$ and $S_U$ differ across directions, the error decomposition depends on how $\Delta$ aligns with the eigenspaces of these respective prompts. The following corollary highlights such behavior.

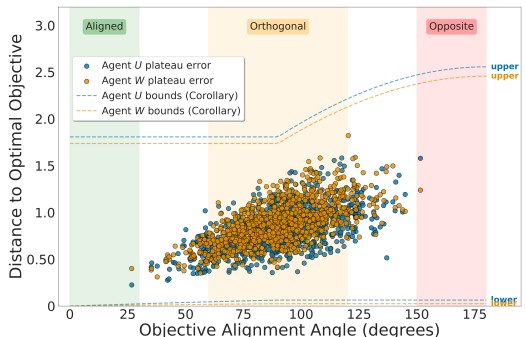

Figure 2: **Plateau error v.s. objective angle** - Plateau error of Agents $W$ (blue) and $U$ (orange) as a function of the objective alignment angle (1000 LSA agent-to-agent interactions). We display the theoretical bounds from Corollary 2 for each agent (lower and upper). As the bounds in Corollary 2 characterize, larger alignment angles correspond to higher plateau errors.

**Corollary 1.** *Assume $S_W$ and $S_U$ commute so they are simultaneously diagonalizable with eigenvalues $\Lambda_W, \Lambda_U$, and let $\tilde{\Delta}$ be the projection of $\Delta$ in their eigenbasis. Then, as $\eta \to 0$,*

$$\|u_\infty - u^\star\|_2^2 = \sum_{i=1}^d \left(\frac{\lambda_{w,i}}{\lambda_{w,i}+\lambda_{u,i}}\right)^2 \tilde{\Delta}_i^2 + O(\eta) \qquad \|w_\infty - w^\star\|_2^2 = \sum_{i=1}^d \left(\frac{\lambda_{u,i}}{\lambda_{w,i}+\lambda_{u,i}}\right)^2 \tilde{\Delta}_i^2 + O(\eta)$$

*(Proof in Appendix 9.3)*

In the commuting case, the misalignment $\Delta$ decomposes into independent spectral directions, and each agent's plateau is obtained by weighting the per-mode discrepancy $\tilde{\Delta}_i$. Along a mode $i$ where $\lambda_{w,i} \gg \lambda_{u,i}$, the $U$ agent error is *amplified* while that of $W$ agent is *suppressed* and vice versa. Thus anisotropy acts as a directional filter: each agent incurs larger errors precisely in the directions where the other agent's geometry dominates.

Now that we understand how the prompt and its induced geometry affects each agent's asymptotic error, we are interested in the impact of objective discrepancy. The following corollary provides a description of the error that each agent will achieve at convergence with respect to the angle between the two objectives.

**Corollary 2.** *Let $S := S_W + S_U$ be invertible, $\theta \in [0, \pi]$ be the angle between $w^\star$ and $u^\star$, then as $\eta \to 0$,*

$$\alpha_U\, r_{\min}(\theta) \;\leq\; \frac{\|u_\infty - u^\star\|_2}{\sqrt{\|w^\star\|_2^2 + \|u^\star\|_2^2}} \;\leq\; \beta_U\, r_{\max}(\theta) \;+\; O(\eta), \tag{5}$$

$$\alpha_W\, r_{\min}(\theta) \;\leq\; \frac{\|w_\infty - w^\star\|_2}{\sqrt{\|w^\star\|_2^2 + \|u^\star\|_2^2}} \;\leq\; \beta_W\, r_{\max}(\theta) \;+\; O(\eta), \tag{6}$$

*where*

$$r_{\min}(\theta) = \min\{1, \sqrt{1-\cos\theta}\}, \qquad r_{\max}(\theta) = \max\{1, \sqrt{1-\cos\theta}\},$$
$$\alpha_U = \sqrt{\lambda_{\min}(S_W S^{-2} S_W)}, \qquad \beta_U = \sqrt{\lambda_{\max}(S_W S^{-2} S_W)},$$
$$\alpha_W = \sqrt{\lambda_{\min}(S_U S^{-2} S_U)}, \qquad \beta_W = \sqrt{\lambda_{\max}(S_U S^{-2} S_U)}.$$

*(Proof in Appendix 9.4)*

From this corollary, the normalized convergence plateaus are *nondecreasing* in $\theta \in [0, \pi]$, bounded between the envelopes $\alpha\, r_{\min}(\theta)$ and $\beta\, r_{\max}(\theta)$ (up to an $O(\eta)$ term), with multiplicative constants $(\alpha_U, \beta_U)$ and $(\alpha_W, \beta_W)$ for agents $U$ and $W$, respectively. This phenomena is observed empirically in Figure 2 where we observe each agent's asymptotic error with respect to the angle between their objective. As formally described in Corollary 2, the plateau error of each agent increases with respect to the angle between their objective.

From Proposition 1, the asymptotic squared errors can be written as $\|u_\infty - u^\star\|_2^2 = \Delta^\top S_W S^{-2} S_W \Delta$, $\|w_\infty - w^\star\|_2^2 = \Delta^\top S_U S^{-2} S_U \Delta$, where $\Delta = u^\star - w^\star$ encodes the discrepancy between the two prompt-induced objectives and $S_W, S_U$ are determined by the prompt geometries. Thus, for fixed prompt geometries, both agents' plateau errors grow quadratically with the size of the objective gap $\|\Delta\|_2$, and are further amplified when $\Delta$ has large components along eigen-directions where $S_W S^{-2} S_W$ or $S_U S^{-2} S_U$ have large eigenvalues. Corollary 2 then shows that, after normalizing by $\|w^\star\|_2^2 + \|u^\star\|_2^2$, these plateau errors are nondecreasing functions of the alignment angle $\theta$ between $w^\star$ and $u^\star$. In our LSA model, both $\Delta$ and the geometries $S_W, S_U$ are fully determined by the prompts (system instructions and in-context examples) given to each LSA agent.

These results yield a concrete prompt-design principle for multi-agent systems: construct the

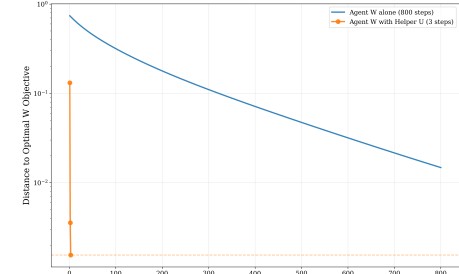

Figure 3: **Cooperative Agents -** We compare the convergence of a single agent $W$ (blue) to the same agent interacting with a cooperative helper $U$ for only 3 alternating steps (orange). The helper's objective is updated dynamically at each turn using only turn-local quantities $(u_t, w_{t+1}, S_W, S_U, \eta)$. Following the analytic construction in Corollary 3, the helper computes a temporary target $u_t^\star = w_{t+1} - [I + (\eta S_U)^{-1}(I - \eta S_U)]z_{t+1}$. This LSA-agents experiment highlights the fact that a helper agent can improve another agent's convergence rate by shaping turn-based objectives.

system and task prompts so that the effective objectives realized in-context are as aligned as possible (small $\|\Delta\|_2$ and small $\theta$), for example by explicitly encoding a shared global objective and avoiding components that pull $w^\star$ and $u^\star$ in different directions.

Up to this point, we have assumed that each agent's objective $(w^\star, u^\star)$ is fixed throughout the interaction. In this *fixed-objective* regime, Proposition 1 and Corollary 2 show that any misalignment inevitably induces nonzero plateaus, so alternating updates cannot improve either agent beyond its single-agent optimum. We now turn to a different regime in which one agent is allowed to *adapt its objective turn-by-turn*: the helper agent $U$ can choose a local target $u_t^\star$ at each interaction step. In this adaptive setting, the same linear dynamics can produce genuinely helpful behavior, with the helper agent accelerating the main agent $W$'s convergence beyond what it is able to achieve itself, instead of degrading it.

We consider the aforementioned alternating agent-to-agent dynamic, but we now considering that the helper agent $U$ can adapt its target $u_t^\star$ over time, that is, the two agent system in Eq. 3 is now defined as,

$$w_{t+1} = u_t - \eta S_W(u_t - w^\star), \qquad u_{t+1} = w_{t+1} - \eta S_U(w_{t+1} - u_t^\star). \tag{7}$$

At the *helper turn* $t$, agent $U$ has access only to the turn-local quantities $(u_t, w_{t+1}, S_W, S_U, \eta)$.

**Corollary 3.** *At the helper turn $t$, define $z_{t+1}$ as the solution of the linear system*

$$S_W z_{t+1} = \left(S_W - \frac{1}{\eta}I\right)(w_{t+1} - u_t),$$

*which depends only on the turn-local quantities $(u_t, w_{t+1}, S_W, \eta)$. If the $U$-agent (helper) chooses its turn-specific target*

$$u_t^\star = w_{t+1} - \left[I + (\eta S_U)^{-1}(I - \eta S_U)\right] z_{t+1},$$

*the helper drives the agent-to-agent system directly to agent $W$ optimum, i.e., $w^\star$. (Proof in Appendix 9.5)*

This corollary shows that, by shaping a turn-based objective, the helper agent can implement a Newton-like update for the main agent's quadratic objective using only turn-based information, yielding a substantial acceleration in convergence (Figure 3). Importantly, the helper does not require privileged knowledge of $w^\star$: it constructs its intermediate target solely from observable turn-local quantities (the current iterate, the previous iterate, and the prompt-induced geometries). This stands in stark contrast to the fixed-objective regime studied earlier, where agent-to–agent interaction inevitably yields biased plateaus. Allowing turn-adaptive objectives transforms the same interaction mechanism into a form of cooperative acceleration. Practically, this suggests that multi-agent LLM systems should be designed so that agents can compute or infer helpful intermediate objectives, such as surrogate losses or predicted error directions, rather than being restricted to static, pre-specified goals. Note that, this turn-local helper design directly echoes the "inter-agent misalignment" failures in MAST (Cemri et al., 2025) instead of each agent pursuing a hidden fixed target, the helper's objective $u_t^\star$ is explicitly conditioned on the main agent's current state, providing exactly the correction that the main agent needs at that step and thereby repairing the breakdown in information flow that MAST associates with collapsed "theory of mind".

In this section, we established explicit expressions for the asymptotic errors of both agents (Proposition 1), showing that convergence plateaus are determined jointly by objective misalignment $\Delta$ and the spectral geometry of the prompts $(S_W, S_U)$. Corollary 2 further explains the monotonic growth of normalized plateau errors with the inter-objective angle, providing a predictive lens on non-cooperative agent-to-agent interactions. Taken together, these results characterize the *fixed-objective* multi-agent regime, where misalignment cannot be corrected during inference and residual errors are unavoidable.

Our cooperative construction (Corollary 3) shows that misalignment need not be inherent: in an *adaptive-objective* regime, a helper agent can update a turn-local objective using only observable quantities to realize Newton-like acceleration for the main agent. This demonstrates that the same interaction interface can yield either misalignment-induced degradation or cooperative convergence gains, depending on whether agent objectives are fixed or allowed to adapt. This dichotomy highlights a practical design principle for LLM-based multi-agent systems: prompt structures that stabilize or align objectives mitigate harmful fixed-point biases, while agents capable of constructing turn-local surrogate objectives can actively enhance one another's optimization dynamics.

# 4 ASYMMETRIC CONVERGENCE AND GEOMETRIC CHARACTERIZATION OF ADVERSARIAL AGENTS

We presently develop a theoretical framework for adversarial agents. Specifically, we first characterize geometric conditions under which *asymmetric convergence* is achievable in an agent-to-agent system. That is, is it possible to tune the interaction, via the choice of prompts, so that one agent converges exactly to its objective, while the other agent does not.

## 4.1 ASYMMETRIC CONVERGENCE CONDITIONS

The following proposition presents conditions on the fixed-point equations of the system to achieve asymmetric convergence.

**Proposition 2.** *Asymmetric convergence (i.e., $u_\infty = u^\star$ but $w_\infty \neq w^\star$) occurs if and only if*

$$\Delta \in \ker\left((I - \eta S_U) S_W\right) \qquad and \qquad \Delta \notin \ker\left(\eta S_W - I\right). \tag{8}$$

*(Proof in Appendix 9.6)*

The first condition in equation 8 says that the part of the objective gap $\Delta = u^\star - w^\star$ that $W$ would try to correct is *nullified* by $U$'s turn: whatever $W$ injects along $\Delta$ through its gradient direction $S_W \Delta$ lands in the nullspace of $(I - \eta S_U)$, so $U$ cancels it and can still steer itself exactly to $u^\star$. The second condition excludes a degenerate "one–step fix" for $W$ (i.e., $\Delta$ lying in the eigenspace of $S_W$ with eigenvalue $1/\eta$), which would otherwise let $W$ also eliminate its residual and remove asymmetry. This reasoning can be obtained by looking at the agent-to-agent composed two-turn agent $U$ update $u_{t+1} = (I - \eta S_U)w_{t+1} + \eta S_U u^\star$, $w_{t+1} = u_t - \eta S_W(u_t - w^\star)$ thus,

$$u_{t+1} = \eta S_U u^\star + (I - \eta S_U)u_t - \eta \underbrace{(I - \eta S_U) S_W (u^\star - w^\star)}_{=\,0 \text{ by Eq. equation 8}} - \eta(I - \eta S_U)S_W(u_t - u^\star).$$

Similarly we can decompose agent $W$ next-step error to obtain

$$w_{t+1} - w^\star = (I - \eta S_W)(u_t - u^\star) + \underbrace{(I - \eta S_W)\Delta}_{\text{misalignment term}}.$$

If $\Delta$ is an eigenvector of $S_W$ with eigenvalue $1/\eta$, then $(I - \eta S_W)\Delta = 0$, so $W$ eliminates its residual along that misalignment direction, therefore undoing the asymmetry.

**Corollary 4.** *If the asymmetric convergence condition is not exactly satisfied, i.e., $(I - \eta S_U)S_W \Delta = r$ with $r \neq 0$ then,*

$$\|u_\infty - u^\star\| \leq \|\left(S_U + (I - \eta S_U)S_W\right)^{-1}\| \, \|r\|.$$

*Moreover,*

$$w_\infty - w^\star = (I - \eta S_W)\Delta - (I - \eta S_W)\left(S_U + (I - \eta S_U)S_W\right)^{-1}r,$$

*(Proof in Appendix 9.7)*

From this corollary, we obtain that $W$'s plateau equals the exact-case value $(I - \eta S_W)\Delta$ up to an $O(\|r\|)$ correction and $U$'s plateau grows linearly with the residual $\|r\|$ in the approximate regime. Therefore the asymmetric convergence decays smoothly as the alignment condition not satisfied.

Now we propose to leverage these conditions to provide a provable asymmetric convergence construction.

**Corollary 5.** *Let $\Delta \neq 0$ and choose $\eta > 0$ such that $(\frac{1}{\eta}, \Delta) \notin \text{spec}(S_W)$. Define $v := S_W \Delta$ and let $P_v$ denote the orthogonal projector onto $\text{span}\{v\}$. Set*

$$S_U = \frac{1}{\eta} P_v + \varepsilon (I - P_v) \quad for\ any\ \varepsilon \in (0, \tfrac{1}{\eta}).$$

*Then the agent-to-agent dynamics exhibit* asymmetric convergence*: agent $U$ reaches its objective while agent $W$ does not. (Proof in Appendix 9.8)*

The construction sets $S_U$ to place an *eigenvalue spike* exactly on the problematic direction $v := S_W \Delta$ and to be near–isotropic elsewhere. Because $S_U v = \frac{1}{\eta}v$, we get

$$(I - \eta S_U)v = 0 \implies (I - \eta S_U)S_W \Delta = 0,$$

which satisfies the kernel criterion in Proposition 2. The small $\varepsilon(I - P_v)$ term makes $S_U$ full–rank for stability while keeping $U$'s behavior essentially unchanged on $\text{span}\{v\}$. The side condition $(\frac{1}{\eta}, \Delta) \notin \text{spec}(S_W)$ prevents a symmetric one–step elimination for $W$.

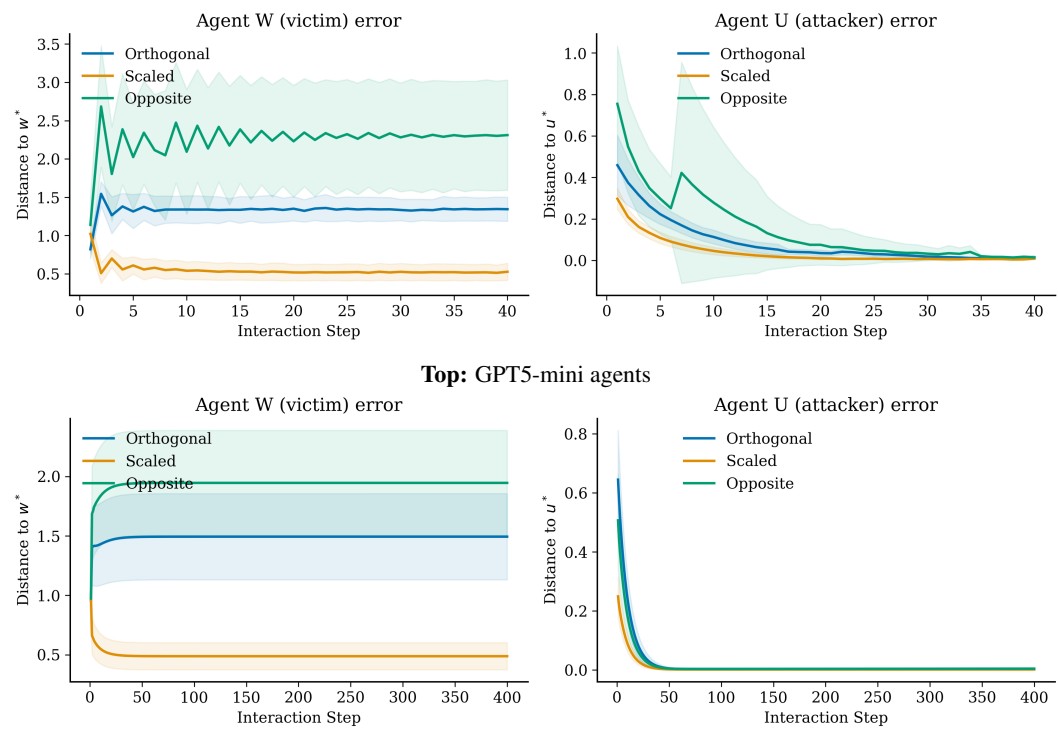

**Top:** GPT5-mini agents

**Bottom:** LSA-trained agents

Figure 4: **White-box agent-to-agent attack.** We evaluate the adversarial algorithm proposed in Algorithm 1 from Section 4 under three objective-gap settings—*orthogonal*, *scaled*, and *opposite*, e.g., opposite is defined as $u^\star = -w^\star$. Each panel plots the *mean* trajectory across 100 runs with shaded $\pm$ std bands (learning rate $\eta = 0.005$). *Left:* distance of the *victim* (Agent $W$) to its target $w^\star$ over interaction steps. In all conditions, $W$ converges to a *nonzero plateau* whose level depends on the gap geometry, as predicted by Proposition 1 and the angle bounds in Corollary 2. *Right:* distance of the *attacker* (Agent $U$) to $u^\star$. Consistent with the kernel criterion $(I-\eta S_U)S_W\Delta = 0, U$ rapidly drives its error to (near-)zero, yielding one-sided success. **Top:** GPT5-mini agents ,early-step variability reflects model decoding the noise but does not alter the outcome. **Bottom:** LSA-trained agents, same protocol; Overall, both agent-base match the theory: anisotropy plus misalignment induces a predictable bias for $W$, while the adversarial spike in $S_U$ yields fast convergence for $U$.

## 4.2 WHITE-BOX AGENT-TO-AGENT ATTACK

In the white-box setting, the adversarial agent has complete knowledge of the target agent's geometry matrix $S_W$ and objective $w^\star$. Note that this scenario is realistic as one can either perform prompt extraction techniques (Zou et al., 2023; Yang et al., 2024; Li et al., 2025) or simply by guessing the other agent prompt and objective prior to the agent-to-agent interaction.

Given knowledge of $(S_W, w^\star, u^\star)$, the attacker's goal is to construct an optimal attack geometry $S_U$ such that the agent-to-agent conversation converges to the attacker's objective $u^\star$ while preventing the victim from reaching $w^\star$. The key insight from Proposition 2 is to design $S_U$ such the part of the gap that $W$ pushes ($S_W\Delta$) falls exactly in the set of directions that $U$ deletes in one step, while the gap itself ($\Delta$) avoids the directions $W$ can delete in one step. Practically, Corollary 5 provides a way to perform such a *white-box attack*. The steps required are as follows: $(i)$ compute $v = S_W\Delta$ (with $P_v := \frac{vv^\top}{\|v\|^2}$), $(ii)$ set $S_U = \frac{1}{\eta}P_v + \varepsilon(I - P_v)$ with small $\varepsilon > 0$. These steps are further described in Algorithm 1.

In Figure 4 we show the empirical result of the white-box attack algorithm described in Algorithm 1 for both the trained LSA agent and GPT5. The resulting dynamics match our theoretical results: the

misalignment drive is canceled by the attacker (agent $U$), yielding fast convergence to $u^\star$, whereas the victim (agent $W$) inherits a persistent bias.

We showed that asymmetric convergence is a geometric feature of the coupled updates: it occurs exactly when the misalignment vector $\Delta$ is annihilated by $(I - \eta S_U)S_W$ yet not by $(I - \eta S_W)$. This yields a constructive recipe, place an eigenvalue spike of $S_U$ on $v = S_W \Delta$ and keep $S_U$ otherwise near-isotropic, so that agent $U$ converges to $u^\star$ while agent $W$ retains a predictable residual.

## 5    EXPERIMENTAL SETTINGS

We now provide the details regarding the experimental results provided throughout the paper. Note that the details for training the LSA model to perform gradient descent update are described in Appendix 8.1 and the algorithms are described in Appendix 6.2.

The inference is based on turn-based interactions between two inference agents $\mathcal{A}_1, \mathcal{A}_2$ that each produce a gradient-like update toward their own linear-regression objective using only in-context information (dataset and shared iterate history). The shared iterate is updated after each agent's call; the next agent receives the updated history $w_{0:t-1}$. This approach is identical for both our LSA-trained agents and our GPT5-based agent and is described in Algorithm 2.

**LSA-trained agents:** For LSA agents, $\mathcal{A}_i$ are a trained single-layer linear self-attention (LSA) model (Section 2) that, at each turn, maps the concatenated tokenized $(X_i, y_i)$ and the iterate history $w_{0:t-1}$ to a gradient-like vector approximating $\nabla L_i(w_t)$, with $L_i(w) = \|X_i^\top w - y_i\|^2$. We evaluate generalization to unseen $(X_i, y_i)$ in the single-agent setting and then use the same checkpoints into Algorithm 2.

**GPT5-based agent:** For the GPT-based agent, we wrap a GPT5 model (gpt-5-mini) in a typed interface that returns a $d$-dimensional gradient given $(X, y, w_t)$ and history $w_{0:t-1}$. Concretely, $\mathcal{A}_{\text{GPT}}$ receives a *system prompt* that explains the objective and formula, and a *user message* containing the exact matrices $X \in \mathbb{R}^{d \times n}$, $y \in \mathbb{R}^n$, the current weight $w_t \in \mathbb{R}^d$, and the history $w_{0:t-1}$. In fact, we are not directly using the $Z$ input as for the LSA agents, we are using its equivalent prompted version defined in Appendix 8.3. Besides, on the output side, the model is constrained to output a float vector as output, i.e., the predicted gradient update. This is performed using a pydantic formatted output schema, also described in Appendix 8.3. Now, the same algorithm as the one defined for the LSA agent is utilized to have the agent-to-agent interactions as defined in Algorithm 2. Additional details about the GPT5 setup and prompt are described in Appendix 8.3.

## 6    CONCLUSION

We introduced a theoretical framework that analyzed multi-agent interactions between LSA-based gradient-descent agents. In the *fixed-objective* regime, where each agent optimizes toward its prompt-induced objective throughout inference, we showed that alternating updates converge to biased fixed points. These residuals are jointly determined by objective misalignment and the anisotropic geometry of agent-specific prompts, yielding explicit, a priori predictions of convergence plateaus and revealing when two agents mutually degrade each other's performance. Within this regime, we further identified conditions under which the dynamics become *asymmetric*, allowing one agent to reach its objective exactly while the other is left with a persistent bias. This leads to constructive mechanisms for adversarial prompt design, where an attacker can suppress or cancel the corrective directions of another agent while preserving its own progress.

We also showed that this behavior is not inherent to multi-agent systems. In an *adaptive-objective* regime, a helper agent can update a turn-based objective using only observable states and prompt-induced geometry. Our construction demonstrates that such a helper can implement Newton-like acceleration for the main agent, transforming the same interaction interface from a source of mutual degradation into a mechanism for cooperative optimization. This highlights a practical design principle for multi-agent LLM systems: when objectives can adapt, agents can construct turn-local surrogate goals that stabilize, align, or accelerate each other's optimization dynamics.

Overall, our results connect the dynamics of in-context gradient descent to the emergent behavior of multi-agent LLM systems, illuminating both the risks of fixed misaligned objectives (plateaus, asymmetries, adversarial vulnerabilities) and the opportunities for principled cooperative acceleration when agents can reshape objectives during interaction.

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

## Supplementary Material

### 6.1 Limitations and scope

Our experiments are restricted to synthetic in-context linear regression and LSA agents, and we only probe GPT-5-mini on the *same* linear-regression tasks. As such, our results do not directly explain the behavior of multi-agent LLM collaborations on open-ended reasoning, writing, or code-generation benchmarks. Instead, we view the present work as a mechanistic case study of two interacting in-context optimizers, which yields concrete, testable hypotheses (e.g., about how representation geometry and objective misalignment interact) that future empirical work on real multi-agent LLM systems can probe.

### 6.2 Algorithms

---

**Algorithm 1** White-box Attack - Prompt Design

---

**Require:** $S_W \in \mathbb{R}^{d \times d}$, mismatch $\Delta = w^\star - u^\star \in \mathbb{R}^d$, stability margin $\tau \in (0, 1/2)$ (e.g. 0.1), step size $\eta$

  **Build the line-space and its projector**

1: Set $v \leftarrow S_W \Delta$.

2: Set $P_v \leftarrow \frac{vv^\top}{\|v\|^2}$ (projector onto $\mathrm{span}\{v\}$).

  **Build the adversarial geometry** $S_U$.

3: Pick any $\varepsilon \in (0, 1/\eta)$ (e.g. $\varepsilon \leftarrow \frac{1-\tau}{\eta}$).

4: Set

$$S_U \leftarrow \frac{1}{\eta} P_v + \varepsilon (I - P_v).$$

  **Realize** $S_U$ **as a data covariance.**

5: Factor $S_U$ as $S_U = LL^\top$.

6: Form $X_\Gamma \in \mathbb{R}^{d \times n}$ with columns spanning $\mathrm{Im}(L)$, e.g. $X_\Gamma \leftarrow \sqrt{n}\, L$.

7: **return** $X_U, \eta$.

---

---

**Algorithm 2** Agent-to-Agent Interaction (Model-agnostic Inference)

---

1: **Inputs:** agents $\mathcal{A}_1, \mathcal{A}_2$; datasets $(X_1, y_1), (X_2, y_2)$; step size $\eta$; max steps $S$

2: $w^{(0)} \leftarrow 0_d$

3: **for** $s = 1$ to $S$ **do**

4:   $\hat{g}_1^{(s)} \leftarrow \mathcal{A}_1\big(X_1, y_1, [w^{(0)}, \ldots, w^{(2s-2)}]\big)$

5:   $w^{(2s-1)} \leftarrow w^{(2s-2)} - \eta \hat{g}_1^{(s)}$

6:   $\hat{g}_2^{(s)} \leftarrow \mathcal{A}_2\big(X_2, y_2, [w^{(0)}, \ldots, w^{(2s-1)}]\big)$

7:   $w^{(2s)} \leftarrow w^{(2s-1)} - \eta \hat{g}_2^{(s)}$

8: **end for**

9: **Return** $w^{(0:2S)}$

---

## 7 Discussion around defense mechanism in Multi-Agent System

Beyond the specific white-box construction we provide, our analysis suggests three general principles for robust multi-agent design:

(i) *Objective alignment.* Since all plateau errors are quadratic in $\Delta = u^\star - w^\star$, the most effective way to reduce vulnerability is to enforce a shared global objective across agents (common system-level task and safety prompt), so that $\|\Delta\|$ is small. This directly shrinks the quadratic forms in Proposition 1 for any attacker geometry.

(ii) *Geometry control.* The strongest instabilities arise when an agent can engineer highly anisotropic prompt geometries $(S_W, S_U)$ with eigenvalues near $1/\eta$ along misalignment directions. Constraining prompt templates to avoid extreme concentration on a single feature direction, and regularizing

estimated geometries toward more isotropic or bounded-spectral forms, keeps $(I - \eta S_U)S_W \Delta$ away from zero and thus prevents large asymmetric plateaus.

(iii) *Interaction protocol.* Our dynamics make explicit how alternating updates propagate misalignment. In practice, one can reduce this channel by limiting how much an untrusted agent can directly overwrite the shared state (e.g., mixing updates with a trusted baseline, or routing critical decisions through oversight agents whose prompts are deliberately aligned). In all cases, the quantities appearing in our theory, $\Delta$, $S_W$, $S_U$, and the residual $r = (I - \eta S_U)S_W \Delta$, can be estimated (e.g., via probing) and used as diagnostics: if they predict large plateau errors, the configuration is structurally fragile and should be revised.

# 8  ADDITIONAL EXPERIMENTAL DETAILS

## 8.1  LSA AGENT TRAINING

For the CoT LSA training, we follow the guidance defined in Huang et al. (2025b). The hyperparameters used for training are defined in Appendix 8.2 Table 8.2.

Each dataset is an i.i.d. linear regression problem of dimension $d$ as defined in Section 2.

$$X \in \mathbb{R}^{d \times n} \sim \mathcal{N}\left(0, \tfrac{1}{d}I\right), \qquad w^\star \sim \mathcal{N}\left(0, \tfrac{1}{d}I\right), \qquad y = X^\top w^\star \in \mathbb{R}^{n \times 1}.$$

From $(X, y)$ we generate a ground truth gradient-descent trajectory on the quadratic loss with learning rate $\eta$. $L(w) = \frac{1}{2}\|X^\top w - y\|_2^2$:

$$g_t = \nabla L(w_t) = X(X^\top w_t - y), \qquad w_{t+1} = w_t - \eta\, g_t, \qquad w_0 = 0.$$

The trajectory is truncated whenever $\|g_t - g_{t-1}\|_2 \le 10^{-3}$ and we retain $\{(w_t, g_t)\}_{t=0}^{\text{max\_iter}}$.

The LSA model is trained to predict the *next gradient descent vector* given all tokens up to the current step. We organize the inputs as a token matrix as defined in Section 2 where the bottom block contains the running weight tokens $w_0, \ldots, w_t$ and a bias row of ones.

Given a dataset and a step index $t \in \{1, \ldots, \text{max\_iter}\}$, we present tokens up to $t - 1$ and regress the next ground truth gradient $g_t$:

$$\mathcal{L}_{\text{step}} = \big\|\text{LSA}(Z_{w_{0:t-1}}) - g_t\big\|_2.$$

We train the LSA with Adam optimizer with learning rate $\eta$ and apply a cosine annealing scheduler.

## 8.2  HYPERPARAMETERS

| Parameter | Default | Description |
|---|---|---|
| d | 10 | data dimension |
| n | 20 | number of in-context examples |
| num_datasets | 100 | independent training datasets |
| batch_size | 512 | (dataset, step) pairs per optimizer step |
| epochs | 100 | passes over the shuffled pair list |
| $\eta$ | 0.005 | step size used to generate GD trajectories |
| scheduler | cosine | $\eta_{\min} = 0.005$ |
| eval datasets | 10 | sampled and averaged per evaluation call |

## 8.3  GPT5 EXPERIMENTAL SETUP

**Model and decoding.**  We use `gpt-5-mini` with JSON-parsed outputs. Unless otherwise noted: temperature $= 0.0$, top_p $= 1.0$, frequency/presence penalties $= 0$, reasoning effort `low`, and a strict response schema (below). Each call is retried up to 3 times on parse/shape failure.

## 8.4  TYPED SCHEMA AND PROMPTS

**Response schema (Pydantic-style)**

```
class GradientResponse(BaseModel):
    thinking: str          # scratchpad text (ignored)
    gradient_next: List[float]  # length d, the gradient \Delta L
```

**System prompt**   The system message provide the objective and dimensionalities for the current dataset ($X \in \mathbb{R}^{d \times n}, y \in \mathbb{R}^n$):

```
You are an expert optimization agent working on linear regression
↪   gradient descent.

        PROBLEM SETUP:
        - Input features X: {d}x{n} matrix (values provided in each
          ↪   request)
        - Target values y: {n}-dimensional vector (values provided in
          ↪   each request)
        - Current weight w: {d}-dimensional vector (what you'll
          ↪   receive)

        TASK: Calculate the gradient \Delta L with respect to w, where
          ↪   L = ||X^T w - y||^2

        FORMULA: \Delta L = X(X^T w - y)
        - X^T w produces an {n}-dimensional vector (predictions)
        - X^T w - y produces an {n}-dimensional vector (residuals)
        - X @ (residuals) produces a {d}-dimensional vector (gradient)

        CRITICAL:
        1. Use the EXACT X and y matrices provided in each request
        2. Your output gradient must be exactly {d}-dimensional
        3. Do NOT make up dummy data - use the actual matrices given
        4. Perform the calculation step by step

  The user will provide w_current and the matrices X, y. Calculate and
    ↪   return the {d}-dimensional gradient vector, do not ask the user to
    ↪   validate what is to be done. The user will not be able to interact
    ↪   with you. Be highly precise and accurate on your computations, you
    ↪   will be evaluated on the distance with the ground truth
    ↪   gradient."""
```

**User message (per turn).**   At turn $t$, we pass the exact numerics for $X, y, w_t$ and the prior history $w_{0:t-1}$. Note that history is included for parity with LSA and to allow in-context, multi-turn conditioning as well as to give the model the capability to perform filtering and negate the attack.

## 9   PROOFS

### 9.1   FIXED POINT ASSUMPTION

**Lemma 1.** *If $S_W, S_U \succ 0$ and*
$$0 < \eta < \min \left\{ \frac{2}{\lambda_{\max}(S_W)}, \ \frac{2}{\lambda_{\max}(S_U)} \right\},$$
*then the fixed point exists and is unique. (Proof in Appendix 9.1)*

*Proof.* For any SPD $S$, the eigenvalues of $M := I - \eta S$ are $\mu_i = 1 - \eta \lambda_i(S)$, so $\|M\|_2 = \max_i |1 - \eta \lambda_i(S)| < 1$ whenever $0 < \eta < 2/\lambda_{\max}(S)$. Thus
$$\rho(M_U M_W) \ \leq \ \|M_U M_W\|_2 \ \leq \ \|M_U\|_2 \|M_W\|_2 \ < \ 1.$$

At a fixed point $(w_\infty, u_\infty)$ we have
$$w_\infty = M_W u_\infty + \eta S_W w^\star,$$
$$u_\infty = M_U w_\infty + \eta S_U u^\star.$$

Eliminating $w_\infty$ from the second equation gives

$$u_\infty = M_U(M_W u_\infty + \eta S_W w^\star) + \eta S_U u^\star = (M_U M_W)u_\infty + \eta(M_U S_W w^\star + S_U u^\star).$$

Equivalently,

$$\left(I - M_U M_W\right) u_\infty = \eta(M_U S_W w^\star + S_U u^\star) =: b. \tag{9}$$

By the step-size assumption we already showed $\|M_U\|_2 < 1$ and $\|M_W\|_2 < 1$, hence

$$\|M_U M_W\|_2 \leq \|M_U\|_2 \|M_W\|_2 < 1,$$

so in particular $\rho(M_U M_W) \leq \|M_U M_W\|_2 < 1$. Therefore $I - M_U M_W$ is invertible and, by the Neumann series,

$$\left(I - M_U M_W\right)^{-1} = \sum_{k=0}^{\infty}(M_U M_W)^k.$$

Applying this inverse to equation 9 yields the unique solution

$$u_\infty = \left(I - M_U M_W\right)^{-1} b = \sum_{k=0}^{\infty}(M_U M_W)^k \, \eta(M_U S_W w^\star + S_U u^\star).$$

Finally,

$$w_\infty = M_W u_\infty + \eta S_W w^\star.$$

Uniqueness follows because $I - M_U M_W$ is nonsingular: if two fixed points give $u_\infty, \tilde{u}_\infty$, then $\left(I - M_U M_W\right)(u_\infty - \tilde{u}_\infty) = 0 \Rightarrow u_\infty = \tilde{u}_\infty$, and the corresponding $w_\infty$ is then uniquely determined by the first line. $\qquad\square$

### 9.2 Proof of Proposition 1

*Proof.*

At convergence (omitting $\infty$ for simplicity), insert equation 2 into equation 3:

$$u = \left[u - \eta S_W(u - w^\star)\right] - \eta S_U\left(\left[u - \eta S_W(u - w^\star)\right] - u^\star\right)$$
$$= u - \eta S_W(u - w^\star) - \eta S_U\left(u - u^\star - \eta S_W(u - w^\star)\right).$$

Subtract $u$ from both sides and factor the terms in $(u - w^\star)$:

$$0 = -\eta S_W(u - w^\star) - \eta S_U(u - u^\star) + \eta^2 S_U S_W(u - w^\star)$$
$$= -\eta\Big[\underbrace{S_W + S_U - \eta S_U S_W}_{\text{matrix}}\Big](u - w^\star) + \eta S_U(u^\star - w^\star).$$

Using $\Delta = u^\star - w^\star$ and canceling $\eta > 0$ gives the linear system

$$\left(S - \eta S_U S_W\right)(u - w^\star) = S_U \Delta. \tag{10}$$

Thus, equation 10 yields

$$u - w^\star = \underbrace{(S - \eta S_U S_W)^{-1} S_U}_{=:H} \Delta.$$

By definition,

$$r_U := u - u^\star = (u - w^\star) - (u^\star - w^\star) = H\Delta - \Delta = -(I - H)\Delta.$$

From equation 2, $w - w^\star = (u - w^\star) - \eta S_W(u - w^\star) = (I - \eta S_W)(u - w^\star) = M_W(u - w^\star)$.

Thus,

$$r_W := w - w^\star = M_W H \Delta, \quad \text{and} \quad r_U = -(I - H)\Delta$$

with $H = (S - \eta S_U S_W)^{-1} S_U$ and $M_W = I - \eta S_W$.

Now,

$$(S - \eta S_U S_W)^{-1} = S^{-1} + \eta S^{-1} S_U S_W S^{-1} + O(\eta^2),$$

thus, $H = S^{-1}S_U + O(\eta)$.

Therefore,

$$r_U = -(I - H)\Delta = -(I - S^{-1}S_U)\Delta + O(\eta) = -S^{-1}S_W\,\Delta + O(\eta),$$

and

$$r_W = (I - \eta S_W)(S^{-1}S_U + O(\eta))\Delta = S^{-1}S_U\,\Delta + O(\eta),$$

Finally, since $S_W^\top = S_W$, $S^{-T} = S^{-1}$, we have

$$\|r_U\|_2^2 = \Delta^\top S_W S^{-2} S_W \Delta + O(\eta), \qquad \|r_W\|_2^2 = \Delta^\top S_U S^{-2} S_U \Delta + O(\eta).$$

$\square$

## 9.3 PROOF OF COROLLARY 1

*Proof.* By Proposition 1,

$$\|u_\infty - u^\star\|_2^2 = \Delta^\top(S_W S^{-2} S_W)\Delta + O(\eta), \qquad \|w_\infty - w^\star\|_2^2 = \Delta^\top(S_U S^{-2} S_U)\Delta + O(\eta),$$

with $S := S_W + S_U$. Assume $S_W$ and $S_U$ commute. Then there exists an orthonormal $Q$ such that

$$S_W = Q\,\mathrm{diag}(\lambda_w)\,Q^\top, \quad S_U = Q\,\mathrm{diag}(\lambda_u)\,Q^\top, \quad S = Q\,\mathrm{diag}(\lambda_w + \lambda_u)\,Q^\top,$$

where $\lambda_{w,i}, \lambda_{u,i} \geq 0$ and $\lambda_{w,i} + \lambda_{u,i} > 0$ for all $i$ since $S$ is invertible. Hence

$$S^{-2} = Q\,\mathrm{diag}\big((\lambda_w + \lambda_u)^{-2}\big)\,Q^\top,$$

and a direct multiplication yields

$$S_W S^{-2} S_W = Q\,\mathrm{diag}\left(\frac{\lambda_w^2}{(\lambda_w + \lambda_u)^2}\right)Q^\top, \qquad S_U S^{-2} S_U = Q\,\mathrm{diag}\left(\frac{\lambda_u^2}{(\lambda_w + \lambda_u)^2}\right)Q^\top.$$

Let $\tilde{\Delta} := Q^\top \Delta$. Substituting into the quadratic forms gives

$$\|u_\infty - u^\star\|_2^2 = \sum_{i=1}^d \left(\frac{\lambda_{w,i}}{\lambda_{w,i} + \lambda_{u,i}}\right)^2 \tilde{\Delta}_i^2 + O(\eta), \qquad \|w_\infty - w^\star\|_2^2 = \sum_{i=1}^d \left(\frac{\lambda_{u,i}}{\lambda_{w,i} + \lambda_{u,i}}\right)^2 \tilde{\Delta}_i^2 + O(\eta),$$

$\square$

## 9.4 PROOF OF COROLLARY 2

*Proof.* From the fixed–point identities (see Proposition 1 and its proof), a Neumann expansion gives

$$r_U := u_\infty - u^\star = -(S^{-1}S_W)\Delta + O(\eta), \qquad r_W := w_\infty - w^\star = (S^{-1}S_U)\Delta + O(\eta),$$

where $S := S_W + S_U$ and $\Delta := u^\star - w^\star$.

$$\|r_U\|_2^2 = \Delta^\top \underbrace{(S_W S^{-2} S_W)}_{=:C_U}\Delta + O(\eta), \qquad \|r_W\|_2^2 = \Delta^\top \underbrace{(S_U S^{-2} S_U)}_{=:C_W}\Delta + O(\eta).$$

For any PSD $K$ and $x$, $\lambda_{\min}(K)\|x\|^2 \leq x^\top K x \leq \lambda_{\max}(K)\|x\|^2$. Apply with $x = \Delta$, $K \in \{C_U, C_W\}$, then take square roots:

$$\sqrt{\lambda_{\min}(C_U)}\,\|\Delta\| \leq \|r_U\| \leq \sqrt{\lambda_{\max}(C_U)}\,\|\Delta\| + O(\eta),$$

and similarly for $W$. Define $\alpha_U := \sqrt{\lambda_{\min}(C_U)}$, $\beta_U := \sqrt{\lambda_{\max}(C_U)}$ (and analogously $\alpha_W, \beta_W$), and divide by $\sqrt{\|w^\star\|^2 + \|u^\star\|^2}$.

Let $m := \|w^\star\|_2$, $g := \|u^\star\|_2$, and $\theta \in [0, \pi]$ be the angle between $w^\star$ and $u^\star$. Now,

$$\|\Delta\|_2^2 = m^2 + g^2 - 2mg\cos\theta.$$

Normalize and set the scale ratio $\rho := g/m$

$$\frac{\|\Delta\|_2}{\sqrt{m^2 + g^2}} = \sqrt{\frac{m^2 + g^2 - 2mg\cos\theta}{m^2 + g^2}} = \sqrt{\frac{1 + \rho^2 - 2\rho\cos\theta}{1 + \rho^2}} =: R_\theta(\rho).$$

To bound this uniformly over $\rho \geq 0$, consider $F(\rho) := R_\theta(\rho)^2 = 1 - \dfrac{2\rho\cos\theta}{1 + \rho^2}$. Then

$$F'(\rho) = -2\cos\theta\,\frac{1 - \rho^2}{(1 + \rho^2)^2}.$$

Now we have

$$\begin{cases} \cos\theta > 0: & F'(\rho) < 0 \text{ for } \rho \in [0, 1),\ F'(\rho) > 0 \text{ for } \rho > 1 \ \Rightarrow\ \rho = 1 \text{ is a global minimum;} \\ \cos\theta < 0: & F'(\rho) > 0 \text{ for } \rho \in [0, 1),\ F'(\rho) < 0 \text{ for } \rho > 1 \ \Rightarrow\ \rho = 1 \text{ is a global maximum;} \\ \cos\theta = 0: & F'(\rho) \equiv 0 \ \Rightarrow\ F(\rho) \equiv 1 \text{ and } R_\theta(\rho) \equiv 1. \end{cases}$$

Evaluate the endpoint limits:

$$\lim_{\rho \to 0} R_\theta(\rho) = \lim_{\rho \to \infty} R_\theta(\rho) = 1, \qquad R_\theta(1) = \sqrt{1 - \cos\theta}.$$

Therefore

$$\min_{\rho \geq 0} R_\theta(\rho) = \min\{1, \sqrt{1 - \cos\theta}\} =: r_{\min}(\theta), \qquad \max_{\rho \geq 0} R_\theta(\rho) = \max\{1, \sqrt{1 - \cos\theta}\} =: r_{\max}(\theta).$$

From (iii) and the bounds in (iv),

$$\alpha_U\, r_{\min}(\theta) \ \leq\ \frac{\|u_\infty - u^\star\|_2}{\sqrt{m^2 + g^2}} \ \leq\ \beta_U\, r_{\max}(\theta) \ +\ O(\eta),$$

and analogously for $W$ with $\alpha_W, \beta_W$. Since $r_{\min}, r_{\max}$ are nondecreasing on $[0, \pi]$ and strictly increasing on $(0, \pi)$, the normalized plateaus grow monotonically with $\theta$ (up to the constants $\alpha, \beta$). $\qquad\square$

## 9.5 Proof of Corollary 3

*Proof.* We recall the cooperative dynamics from Eq. 7:

$$w_{t+1} = u_t - \eta S_W(u_t - w^\star), \qquad u_{t+1} = w_{t+1} - \eta S_U(w_{t+1} - u_t^\star). \tag{11}$$

Define the error of the $W$-agent after its update as

$$e_{t+1} := w_{t+1} - w^\star.$$

From equation 11 we have,

$$S_W w^\star = \tfrac{1}{\eta}\big(w_{t+1} - (I - \eta S_W)u_t\big) = \tfrac{1}{\eta}(w_{t+1} - u_t) + S_W u_t. \tag{12}$$

Subtracting $S_W w^\star$ from $S_W w_{t+1}$ gives

$$\begin{aligned} S_W e_{t+1} &= S_W(w_{t+1} - w^\star) \\ &= S_W w_{t+1} - \Big[\tfrac{1}{\eta}(w_{t+1} - u_t) + S_W u_t\Big] \quad \text{(by equation 12)} \\ &= \Big(S_W - \tfrac{1}{\eta}I\Big)(w_{t+1} - u_t). \end{aligned} \tag{13}$$

Thus the error $e_{t+1}$ satisfies the linear system

$$S_W e_{t+1} = \Big(S_W - \tfrac{1}{\eta}I\Big)(w_{t+1} - u_t).$$

In Corollary 3 we defined $z_{t+1}$ as the unique solution of the same system,

$$S_W z_{t+1} = \Big(S_W - \tfrac{1}{\eta}I\Big)(w_{t+1} - u_t).$$

Since $S_W \succ 0$, this solution is unique, and comparing with equation 13 we obtain

$$z_{t+1} = e_{t+1} = w_{t+1} - w^\star. \tag{14}$$

We now show that the choice of $u_t^\star$ in Corollary 3 forces the realized helper state to be $u_{t+1} = w_{t+1} - z_{t+1}$ (and hence $u_{t+1} = w^\star$).

Let $M_U := I - \eta S_U$ for, the helper target is chosen as

$$u_t^\star = w_{t+1} - \left[ I + (\eta S_U)^{-1} M_U \right] z_{t+1}. \tag{15}$$

Plugging this into the helper update in equation 11 gives

$$u_{t+1} = w_{t+1} - \eta S_U \big( w_{t+1} - u_t^\star \big)$$
$$= w_{t+1} - \eta S_U \Big( w_{t+1} - w_{t+1} + \left[ I + (\eta S_U)^{-1} M_U \right] z_{t+1} \Big)$$
$$= w_{t+1} - \eta S_U \left[ I + (\eta S_U)^{-1} M_U \right] z_{t+1}.$$

Using $M_U = I - \eta S_U$, we have

$$\eta S_U \left[ I + (\eta S_U)^{-1} M_U \right] = \eta S_U + M_U = \eta S_U + (I - \eta S_U) = I.$$

Therefore

$$u_{t+1} = w_{t+1} - z_{t+1}. \tag{16}$$

Combining equation 14 and equation 16 yields

$$u_{t+1} = w_{t+1} - e_{t+1} = w_{t+1} - (w_{t+1} - w^\star) = w^\star,$$

$\square$

### 9.6 PROOF OF PROPOSITION 2

*Proof.* Recall the agent-to-agent fixed-point system

$$w_\infty = M_W u_\infty + \eta S_W w^\star, \qquad u_\infty = M_U w_\infty + \eta S_U u^\star, \tag{17}$$

with $M_W := I - \eta S_W$ and $M_U := I - \eta S_U$. Now, assume $u^\star = u_\infty$, from the first fixed-point equation,

$$w_\infty = M_W u^\star + \eta S_W w^\star = (I - \eta S_W) u^\star + \eta S_W (u^\star + \Delta) = u^\star + \eta S_W \Delta.$$

Substitute $w_\infty$ and $u_\infty = u^\star$ into the second equation of equation 17:

$$u^\star = M_U \big( u^\star + \eta S_W \Delta \big) + \eta S_U u^\star = u^\star + \eta (I - \eta S_U) S_W \Delta,$$

which is equivalent to $(I - \eta S_U) S_W \Delta = 0$, establishing the first condition.

The residual for agent $W$ is

$$w_\infty - w^\star = \big( u^\star + \eta S_W \Delta \big) - (u^\star + \Delta) = (\eta S_W - I) \Delta,$$

so $w_\infty \neq w^\star$ iff $(\eta S_W - I)\Delta \neq 0$, the second condition.

By contraction, the fixed point is unique, hence the iterates converge to $(w_\infty, u_\infty)$ with $u_\infty = u^\star$ and $w_\infty \neq w^\star$. $\square$

### 9.7 PROOF OF COROLLARY 4

*Proof.*
$$u_{t+1} = (I - \eta S_U) w_{t+1} + \eta S_U u^\star, \qquad w_{t+1} = u_t - \eta S_W (u_t - w^\star).$$

Expand $u_{t+1}$:

$$u_{t+1} = \eta S_U u^\star + (I - \eta S_U)[u_t - \eta S_W (u_t - w^\star)]$$
$$= \eta S_U u^\star + (I - \eta S_U) u_t - \eta (I - \eta S_U) S_W (u_t - u^\star) - \eta (I - \eta S_U) S_W (u^\star - w^\star).$$

Let $e_t := u_t - u^\star$ and $r := (I - \eta S_U)S_W\Delta$ with $\Delta := u^\star - w^\star$. Then

$$e_{t+1} = \big[I - \eta\big(S_U + (I - \eta S_U)S_W\big)\big]e_t \ - \ \eta r =: Ae_t - \eta r.$$

At a fixed point $e_\infty$ of the affine recursion we have

$$e_\infty \ = \ Ae_\infty - \eta r \quad \Longleftrightarrow \quad (I - A)\,e_\infty \ = \ -\eta r.$$

Since $I - A = \eta\big(S_U + (I - \eta S_U)S_W\big)$, we obtain

$$e_\infty = -\big(S_U + (I - \eta S_U)S_W\big)^{-1}r,$$

and hence

$$\|u_\infty - u^\star\| = \|e_\infty\| \le \big\|\big(S_U + (I - \eta S_U)S_W\big)^{-1}\big\| \, \|r\|.$$

For $W$, we have

$$w_{t+1} - w^\star = (I - \eta S_W)(u_t - w^\star) = (I - \eta S_W)e_t + (I - \eta S_W)\Delta.$$

Taking $t \to \infty$ gives

$$w_\infty - w^\star = (I - \eta S_W)\Delta + (I - \eta S_W)e_\infty = (I - \eta S_W)\Delta - (I - \eta S_W)\big(S_U + (I - \eta S_U)S_W\big)^{-1}r.$$

$\qquad\square$

### 9.8 PROOF OF COROLLARY 5

*Proof.* Let $S_U \ = \ \frac{1}{\eta}\,P_v \ + \ \varepsilon\,(I - P_v)$ where $v = S_W\Delta$. Now,

$$S_U v \ = \ \tfrac{1}{\eta}v \quad \Longrightarrow \quad (\eta S_U - I)v = 0.$$

thus, $(I - \eta S_U)S_W\Delta = 0$. For $z \in \mathrm{span}(I - P_v)$, $P_v z = 0$ and $(I - P_v)z = z$, hence

$$S_U z \ = \ \varepsilon z \quad \Longrightarrow \quad (\eta S_U - I)z = (\eta\varepsilon - 1)z \ne 0$$

because $\eta\varepsilon - 1 < 0$, thus $S_U$ is full rank.

Now, by assumption $(\frac{1}{\eta}, \Delta) \notin \mathrm{spec}(S_W)$ hence

$$\Delta \notin \ker(I - \eta S_W).$$

Since, $\lambda_{\max}(S_U) = \max\{1/\eta, \, \varepsilon\} = 1/\eta$, so the condition $\eta < 2/\lambda_{\max}(S_U)$ gives $\eta < 2\eta$, trivially verified for $\eta > 0$. $\qquad\square$

## 10 AI TOOLING USAGE

For this paper, AI tools were used for specific purpose: $(i)$ polish writing & code, $(ii)$ search for references, $(iii)$ sounding board for some theoretical results.

