# OpenReview forum: "Convergence dynamics of Agent-to-Agent Interactions with Misaligned objectives"
_ICLR.cc/2026/Conference — ICLR 2026 Conference Desk Rejected Submission_

### Official Review · Reviewer_dPdy · 2025-10-16

**Soundness:** 3
**Presentation:** 3
**Contribution:** 2
**Rating:** 4
**Confidence:** 3

**Summary:**

This paper presents a theoretical framework to analyze the interaction dynamics between two language model-based agents.  It models the agent-to-agent interaction as an alternating optimization process, where each agent performs an in-context gradient update towards its own, potentially misaligned, objective.  The authors provide a formal characterization of the convergence dynamics, showing that misaligned objectives result in a biased equilibrium where neither agent fully achieves its goal.  The framework predicts these residual errors based on the objective gap and the geometry induced by each agent's prompt.  Furthermore, the paper establishes the conditions for asymmetric convergence and proposes a constructive, white-box adversarial attack that allows one agent to achieve its objective while forcing the other to retain a persistent error.  These theoretical results are validated with experiments on in-context linear regression tasks using both trained transformer models and GPT-5.

**Strengths:**

1. The paper offers a clean and analytically grounded model of in-context optimization between interacting agents.

2. The theoretical results (Propositions 1–3) are mathematically sound and provide clear geometric intuition for asymmetric convergence.

3. The analysis extends the “transformers-as-optimizers” view to a two-agent setting, which is conceptually novel and well aligned with the learning theory track.

**Weaknesses:**

1. The paper’s core theory assumes transformers implement in-context gradient-like updates (the “transformers-as-optimizers” view) and then analyzes coupled update dynamics under that assumption. However, the GPT-5 experiments do not test emergent in-context optimization — they prompt the model with the explicit gradient formula and treat GPT-5 as an arithmetic oracle. This weakens the experimental link to the paper’s foundational claim: the GPT-5 results demonstrate correct formula execution, not that modern LLMs naturally realize the assumed optimizer dynamics.

2. Figure 3 reports attack success rates using thresholds ε₁ and ε₂, but the manuscript never specifies these threshold values or how they were chosen. Without concrete ε values (or sensitivity analysis), the reported percentages are uninterpretable and non-reproducible: it is impossible to judge whether “85% success” reflects algorithmic failure modes, threshold arbitrariness, or genuine instability in the dynamics.

3. Algorithm 1 relies on a carefully aligned eigen-spike construction, but the experiments do not compare this design to simpler baselines (e.g., misaligned anisotropy, scalar scaling of S_U, or random high-anisotropy prompts).  As presented, it is unclear whether the geometric construction is uniquely required for one-sided convergence or merely one of many ways to produce non-symmetric outcomes.

**Questions:**

1. The theoretical analysis assumes each agent performs a linear gradient-like update, following the LSA approximation. How robust are the main results, especially Proposition 1 and Corollary 2, when the agent dynamics include moderate nonlinearities or higher-layer effects? It would be valuable to see whether asymmetric convergence still appears under more realistic conditions.

2. Proposition 2 and Corollary 3 rely on exact equalities such as ((I - \eta S_U)S_W \Delta = 0). In practice, these conditions can only be approximately met. How sensitive is the observed asymmetric convergence to small violations of this condition? Do the effects decay smoothly, or is the phenomenon brittle?

3. In the GPT-5 setup, the model is prompted with the exact gradient formula, so it is effectively executing a prescribed computation rather than demonstrating emergent optimization. Could you test a variant where GPT-5 infers the update rule from examples without seeing the explicit formula?

---

> ### Author Response · Authors · 2025-11-21
> **Response to reviewer dPdy (Part 1)**
>
> We sincerely thank the reviewer for the insightful and constructive feedback. We appreciate your careful reading of our work and your helpful suggestions for improving the paper.
>
> **W1 - The paper’s core theory assumes transformers implement in-context gradient-like updates (the “transformers-as-optimizers” view) and then analyzes coupled update dynamics under that assumption. However, the GPT-5 experiments do not test emergent in-context optimization — they prompt the model with the explicit gradient formula and treat GPT-5 as an arithmetic oracle. This weakens the experimental link to the paper’s foundational claim: the GPT-5 results demonstrate correct formula execution, not that modern LLMs naturally realize the assumed optimizer dynamics.**
>
> The goal of the GPT-5 experiment is not to show that LLMs implement in-context gradient updates, in fact that is already proven and covered in the in-context learning literature we provide in the introduction, in fact, a lot of papers have been studying this aspect. In this paper, the goal of the GPT-5-mini experiment is to show that even such a model is not capable of countering the attack albeit having access to the full trace of the gradients and the target samples. Therefore complementing our LSA agent experiments and showing the potential threats of agent-to-agent conversations. More details on this point in the answer to the third question.
>
>
> **W2 - Figure 3 reports attack success rates using thresholds ε₁ and ε₂, but the manuscript never specifies these threshold values or how they were chosen. Without concrete ε values (or sensitivity analysis), the reported percentages are uninterpretable and non-reproducible: it is impossible to judge whether “85% success” reflects algorithmic failure modes, threshold arbitrariness, or genuine instability in the dynamics.**
>
> We appreciate the reviewer’s observation and we agree. After revisiting the analysis, we realized that reporting a binary success rate was misleading (and the thresholds used were different): it obscures the actual phenomenon of interest, which is the structure of the attack trajectories and the induced errors in both the attacker and the victim. For this reason, the updated version of the paper removes the success-rate measure and focuses on the continuous error trajectories, which provide a much clearer and model-agnostic comparison between LSA and GPT-5-mini.
>
> **W3 - Algorithm 1 relies on a carefully aligned eigen-spike construction, but the experiments do not compare this design to simpler baselines (e.g., misaligned anisotropy, scalar scaling of S_U, or random high-anisotropy prompts). As presented, it is unclear whether the geometric construction is uniquely required for one-sided convergence or merely one of many ways to produce non-symmetric outcomes.**
>
> Proposition 2 is an equivalence relationship (if and only if), therefore it is theoretically proven that the condition in proposition 2 is the unique way to achieve asymmetric convergence.

---

> ### Author Response · Authors · 2025-11-21
> **Response to reviewer dPdy (Part 2)**
>
> **Q1 - The theoretical analysis assumes each agent performs a linear gradient-like update, following the LSA approximation. How robust are the main results, especially Proposition 1 and Corollary 2, when the agent dynamics include moderate nonlinearities or higher-layer effects? It would be valuable to see whether asymmetric convergence still appears under more realistic conditions.**
>
> It is important to note that the linear update in our analysis does not arise from any architectural simplification, but from the structure of the task itself. For a quadratic loss, the gradient-descent update is
> inherently affine, i.e., $x_{t+1} = A x_t + b$, and this remains the effective rule implemented in-context by any model, linear or deep, that is trained to perform gradient descent on such tasks as per the related work that can be found in the introduction. If the reviewer has the time, we invite him to check “Transformers Learn to Implement Multi-step Gradient Descent with Chain of Thought” (Huang et al. 2025) or the other related work mentioned in the introduction to better understand the origin of the gradient descent considerations we are making in this paper.
>
>
> **Q2 - Proposition 2 and Corollary 3 rely on exact equalities such as $((I - \\eta S_U)S_W \\Delta = 0)$. In practice, these conditions can only be approximately met. How sensitive is the observed asymmetric convergence to small violations of this condition? Do the effects decay smoothly, or is the phenomenon brittle?**
>
> We thank the reviewer for pointing out the robustness of the result, it is an important aspect that the paper was missing. We added Corollary 4 in the revised paper. The corollary shows how the plateaus vary when the asymmetry conditions are not exactly met. Specifically, we obtain that:
> a)  the $W$’s plateau equals the exact-case value $(I-\\eta S_W)\\Delta$ up to an $O(\\|r\\|)$ correction.
> b)  $U$’s plateau grows linearly with the residual $\\|r\\|$ in the approximate regime.
>
> We can now conclude that the asymmetric convergence decays smoothly as the alignment
> condition is not satisfied exactly. Also, it is important to point out that GPT-5-mini experiment actually represents an agent that behaves approximately like a gradient-based in-context optimizer, yet the adversarial prompt geometries based on the asymmetric convergence theory we develop shows its effect similarly to the LSA case.
>
>
> **Q3 - In the GPT-5 setup, the model is prompted with the exact gradient formula, so it is effectively executing a prescribed computation rather than demonstrating emergent optimization. Could you test a variant where GPT-5 infers the update rule from examples without seeing the explicit formula?**
>
> We agree that in our GPT-5-mini setup the model is given the exact gradient formula. However, the success of the white-box attack is still *not* trivial. A large LLM could in principle disrupt the attack in several ways: by smoothing extreme coordinates, averaging over the weight-history (that we provide in the prompt), or implicitly damping updates that look unstable or ill-conditioned. Any of these behaviors would weaken the kernel condition $(I-\\eta S_U)S_W\\Delta \\approx 0$ that our construction relies on.
> Empirically, GPT-5-mini nonetheless follows the adversarial geometry closely enough that the asymmetric convergence predicted by our theory still appears. This is exactly what we aim to test: *given* an agent that can compute (or approximate) gradient updates, does an adversarial prompt geometry still control the multi-agent dynamics? Our results indicate yes. As such, this experiment is there to highlight potential threats of agent-to-agent conversations even using large LLMs than re-establishing whether or not a large LLM can approximate a gradient of a least square problem.
> The experiment proposed by the reviewer would test *whether* transformers can implement gradient-like updates (already discussed by many papers cited in the introduction), not *how* those updates behave under adversarial multi-agent interaction, which is our focus.
>
> We are eager to hear about your further comments. Please let us know if there is anything else we can clarify.

---

> > ### Comment · Reviewer_dPdy · 2025-11-21
> > **Thank you for the response**
> >
> > Thank you for your detailed responses to my comments. Your explanations have deepened my understanding of the paper's scope and the rigor of your theoretical framework, particularly regarding the linearity assumption for LSA models and the specific goals of your GPT-5 experiments. Actually, I am particularly interested in LLM-based MAS, and I've been closely following recent work on multi-agent failure attribution [1, 2]. I noticed that you mentioned your insights on prompt design in the text. I would like to further ask the author for your insights on how to combine the theories with the construction of practical and robust multi-agent systems.
> >
> > Overall, I feel most of my questions have been well addressed, and I look forward to your further insights. I will consider adjusting my score accordingly.
> >
> > [1] Kong, Fanqi, et al. "Aegis: Automated Error Generation and Attribution for Multi-Agent Systems." arXiv preprint arXiv:2509.14295 (2025).
> >
> > [2] Cemri, Mert, et al. "Why do multi-agent llm systems fail?." arXiv preprint arXiv:2503.13657 (2025).

---

> > > ### Author Response · Authors · 2025-11-21
> > > **Additional response reviewer dPdy  (part 1)**
> > >
> > > We thank the reviewer for helping us to connect our approach to real system design and concrete actionables which is usually one of the bottlenecks of theoretical papers.
> > >
> > > Conceptually, Aegis and MAST are bottom-up tools: they provide datasets, taxonomies, and models for observing and attributing MAS failures across diverse real systems. Our work is top-down: it gives a mechanistic toy model where we can derive closed-form expressions for how misalignment ($\\Delta$) and geometry ($S_W, S_U$) jointly produce biased equilibria, asymmetric convergence, and adversarial vulnerabilities. We also want to highlight the fact that both these works are working on collaborative multi-agent systems, while we also consider adversarial systems.
> > >
> > > **Connection to FC1 (system design issues) in MAST**
> > > MAST identifies *system design issues*, including flaws in agent
> > > roles, and poor or ambiguous prompt instructions, as a primary
> > > source of MAS failures. Our framework provides a mechanistic explanation of this empirical finding. In our model, each agent's prompt induces an implicit quadratic objective $(w^\\star,u^\\star)$
> > > and a prompt geometry $(S_W,S_U)$. When system prompts and role instructions
> > > are not coherently designed, the induced objectives become misaligned
> > > ($\\Delta = u^\\star - w^\\star \\neq 0$). Proposition 1 then shows that, even if
> > > both agents perform ideal in-context gradient descent on their respective
> > > prompts, the alternating interaction necessarily converges to biased plateaus
> > > where neither agent reaches its target. This formalizes MAST's Insight 1:
> > > MAS failure is not only a property of the base model, but also of how prompts
> > > and roles are specified, in our analysis, poor or ambiguous specifications
> > > manifest directly as objective misalignment and hence structural performance
> > > degradation in agent-to-agent dynamics.
> > >
> > > **Connection to FC2 (inter-agent misalignment) in MAST**
> > > MAST identifies FC2 failures as cases where agents proceed under incompatible
> > > assumptions, ignore crucial information from one another, or derail the task
> > > because they fail to infer what information the other agent needs at that
> > > moment, a collapse of “theory of mind” that persists even when communication
> > > channels are available. These failures arise not from missing messages but from
> > > *misaligned internal objectives* that cause agents to update in ways that
> > > are uninformative or even counterproductive for their partner.
> > >
> > > Our fixed-objective dynamics provide an explicit mechanism for this understanding behavior.
> > > When each agent optimizes its own target $(w^\\star,u^\\star)$, Proposition~1
> > > shows that their alternating updates necessarily interfere: each step partially
> > > undoes or distorts the other’s progress, leading to biased plateaus. From the
> > > perspective of MAST, this looks exactly like agents failing to use or supply the
> > > information the other agent needs, even though the “messages’’ themselves are
> > > perfectly readable. To connect the language of the two papers, note that a “collapse of a theory of mind” might be more simply stated as “not considering the other participant” - this is precisely what unfolds when agents have a fixed personal objective throughout the course of conversation.
> > >
> > > Corollary~3 shows how to remove this failure mode. Instead of pursuing a fixed
> > > $u^\\star$, the helper agent $U$ computes a *turn-local objective*
> > > $$
> > > u_t^\\star(u_t, w_{t+1}, S_W, S_U, \\eta)
> > > $$
> > > chosen so that the realized update
> > > $$
> > > u_{t+1} = w_{t+1} - \\eta S_U (w_{t+1} - u_t^\\star)
> > > $$
> > > implements a Newton-like correction for the *main* agent. In effect, the
> > > helper's message is no longer driven by its own objective; it is computed
> > > *solely from the main agent’s current state*. This ensures:
> > >
> > > - the helper no longer injects information that conflicts with the main
> > >   agent’s goal;
> > > - the information it provides is exactly the right piece of information the main agent needs at
> > >   that turn (a curvature-corrected update direction);
> > > - the interaction no longer suffers from the FC2 breakdown in information
> > >   flow, the helper is explicitly conditioned on the main agent’s informational
> > >   requirements.
> > >
> > > Thus, our adaptive regime provides a concrete mathematical instance of the kind
> > > of “improved message content’’ and “reasoning about another agent’s
> > > informational needs’’ that MAST argues are required to resolve FC2 failures.

---

> > > ### Author Response · Authors · 2025-11-21
> > > **Additional response reviewer dPdy (part 2)**
> > >
> > > **Connection to Aegis (Kong et al., 2025).**
> > > Aegis generates large-scale perturbation datasets by injecting controlled
> > > errors into intermediate agent messages and reasoning traces, then observing
> > > how these perturbations propagate through a multi-agent system. In our
> > > framework, such perturbations correspond most directly to changes in the
> > > *state* $(w_t,u_t)$ that an agent receives from its partner at each turn.
> > > These state-level disturbances do not modify the underlying objectives
> > > $(w^\\star,u^\\star)$ or geometries $(S_W,S_U)$, and therefore they do not alter
> > > the closed-form fixed point characterized in Proposition 1. However, when the
> > > agents’ objectives are misaligned ($\\Delta\\neq 0$) and the prompt geometries
> > > are anisotropic, perturbations injected into $(w_t,u_t)$, even when
> > > small and local, are amplified at each update along these biased directions (induced by prompt anisotropy) and can lead to persistent, systematic deviations in the trajectory before convergence. This mirrors the empirical failures catalogued by Aegis: seemingly local message
> > > perturbations cause multi-step cascades of degradation. Our theoretical
> > > framework therefore complements Aegis by explaining why such cascades arise
> > > and by identifying the geometric and misalignment factors that govern their
> > > severity.
> > >
> > > **We did our best to update the paper accordingly and added in the revised version of the manuscript:**
> > >
> > > We added Kong et al. Aegis citation.
> > >
> > > In Sec.3 after Prop.1
> > >
> > > “Recent empirical taxonomies of multi-agent failures, such as MAST, identify *system design issues*, including flawed role specifications and ambiguous prompts, as a primary source of breakdown; in our framework, these design choices manifest as misaligned effective objectives $(w^\\star,u^\\star)$ and ill-designed geometries $(S_W,S_U)$, and Proposition 1 shows that such misalignment inevitably induces biased plateaus even when each agent is individually competent.”
> > >
> > > In Sec.3 after Cor.3
> > >
> > > “ Note that, this turn-local helper design directly echoes the “inter-agent misalignment” failures in MAST instead of each agent pursuing a hidden fixed target, the helper's objective $u_t^\\star$ is explicitly conditioned on the main agent's current state, providing exactly the correction that the main agent needs at that step and thereby repairing the breakdown in information flow that MAST associates with collapsed “theory of mind”.”

---

> > > > ### Comment · Reviewer_dPdy · 2025-11-27
> > > > **Thanks for your response**
> > > >
> > > > Thank you for the detailed response. I believe my main concern has been addressed. I hope the authors can strengthen the connection between the theoretical analysis and practical applications in the updated version of the paper, as this would further enhance its impact and quality. I have raised my score to reflect the same.

---

> > > > > ### Author Response · Authors · 2025-11-28
> > > > > **Response to reviewer dPdy**
> > > > >
> > > > > We are happy to have addressed your main concerns regarding our paper. We thank you for your constructive and thoughtful suggestions and we appreciate your efforts in helping us improve the quality of our paper.

---

### Official Review · Reviewer_d9dT · 2025-10-29

**Soundness:** 3
**Presentation:** 3
**Contribution:** 2
**Rating:** 4
**Confidence:** 3

**Summary:**

This paper theoretically studies agent-to-agent interactions, based on the in-context linear regression task, where both agents are driven by llm and may have misaligned objectives. The authors extend the previous work and formulation on single agent, during the interaction, agents take turns to conduct approximately linear gradient updates towards their own goal using the output of each other as context or input. Utilizing fixed point theory and spectral analysis, the authors analyze how the prompt design and objective difference contribute to the biased equilibrium. They also identify conditions that lead to asymmetric convergence where only one side of the agent reaches its goal, and design a white-box attack algorithm accordingly. Experiments with pretrained single-layered linear self-attention transformer and gpt5 demonstrate the theoretical results and provides insights on understanding llm-based multi-agent systems.

**Strengths:**

First of all, the authors identify an important and timely problem in the field of multi-agent systems (MAS) involving large language models (LLMs). The unpredictability of LLM driven MAS and their occasional under performance compared to single-agent systems highlight the need for a deeper understanding of agent interactions. The paper's focus on characterizing agent-to-agent interactions and the analysis regarding the internal state updates is novel and addresses a gap in the literature.

The theoretical analysis is rigorous and well-supported by mathematical proofs. The inclusion of both LSA and gpt5 agents in the experiments strengthens the credibility of the results. Furthermore, it is nice that the authors link findings on asymmetric conditions to adversarial prompt design and white-box attacks, demonstrating the potential for malicious exploitation and also opens up important discussions on LLM safety.

Regarding the presentation, the paper is well-written and clearly structured. The authors provide comprehensive explanations of the theoretical framework and detailed derivations of key results. Detailed explanation after key results allows readers to easily follow and understand the findings.

**Weaknesses:**

The paper discusses white-box attacks but does not delve into potential defenses. It would be beneficial to add a short discussion regarding the strategies for eliminating or mitigating these attacks. Addressing these concerns would provide a more comprehensive understanding of the security implications and offer practical solutions for securing multi-agent systems.

At the end of section 3, the suggestion to design a common goal for multi-LLMs is quite intuitive. It would be a lot better if the authors could further investigate this idea to provide more valuable insights and practical guidance. For example, they could explore specific techniques for aligning objectives or designing prompts that promote collaboration. This would enhance the paper's contribution to the development of effective multi-agent systems.

**Questions:**

- In section 3, we can derive the plateau levels if given u* and w*. But, if we already know u* w*, what is the point of using multiple llm agents to interact and find the solution?

- In Figure 3, it seems that the victim error in LSA-trained agents are high enough, while the attacker’s error is quite low, then, how come the attack success rate is lower than that in GPT5? If this is correct, what are the possible causes? Is it due to differences in model complexity, training data, or other factors? A deeper analysis would enhance the understanding of the results.

- The white-box attack may cause safety issues, are there any solutions or defense approaches? How to secure the multi-agent system or make them more robust? Further, can we derive some insights on what kind of work is suitable for MAS instead of single-agent from this paper?

---

> ### Author Response · Authors · 2025-11-21
> **Response to reviewer d9dT (Part 1)**
>
> We sincerely thank the reviewer for the insightful and constructive feedback. We appreciate your careful reading of our work and your helpful suggestions for improving the paper.
>
> **W1 - The paper discusses white-box attacks but does not delve into potential defenses. It would be beneficial to add a short discussion regarding the strategies for eliminating or mitigating these attacks. Addressing these concerns would provide a more comprehensive understanding of the security implications and offer practical solutions for securing multi-agent systems.**
>
> We agree with the reviewer that the potential defense mechanisms are critical for agent-to-agent systems. We answer this weakness in detail in the question section below and we edit the paper to try to cover this topic as much as possible.
>
> **W2 - At the end of section 3, the suggestion to design a common goal for multi-LLMs is quite intuitive. It would be a lot better if the authors could further investigate this idea to provide more valuable insights and practical guidance. For example, they could explore specific techniques for aligning objectives or designing prompts that promote collaboration. This would enhance the paper's contribution to the development of effective multi-agent systems.**
>
> This is a great point, and one that other reviewers raised. In the revised
> manuscript we now make this connection explicit by adding a dedicated
> *adaptive multi-agent* subsection and a new theoretical result
> (Corollary 3). In contrast to the fixed-objective regime already analyzed,
> Corollary 3 shows that a helper agent $U$ can *dynamically* set a
> turn-based target $u_t^\\star$, computed solely from observable quantities
> $(u_t, w_{t+1}, S_W, S_U, \\eta)$, such that its realized update implements a
> Newton-like step for the main agent. This requires no access to $w^\\star$ and
> no global coordination, only the turn-by-turn interaction state.
>
> We also include a new experiment (Figure 3) showing that this adaptive
> construction yields a *dramatic acceleration* in convergence compared to a
> single agent running alone. Together, these additions give a concrete
> realization of how multi-agent systems can be beneficial: prompts can be
> designed so that some agents act as *helpers* and infers local surrogate objectives, error directions, or consistency checks that improve another agent’s update.
>
> This clarifies the prompt-design principle referenced in the original
> sentence: when cooperation is needed, prompts should explicitly structure
> roles such that helper agents synthesize intermediate targets from turn-based
> information, turning the same agent-to-agent interface from one that
> manifests misalignment into one that enables cooperative acceleration.

---

> ### Author Response · Authors · 2025-11-21
> **Response to reviewer d9dT (Part 2)**
>
> **Q1 – In section 3, we can derive the plateau levels if given $u^\*$ and $w^\*$. But, if we already know $u^\\star$ and $w^\\star$, what is the point of using multiple LLM agents to interact and find the solution?**
>
> Thank you for this question. We believe the issue stems from a misunderstanding of the role played by $u^\\star$ and $w^\\star$ in our analysis. These quantities are *not* provided to the agents, nor used by them at inference time. They are the (unknown) minimizers of each agent’s underlying quadratic objective, the regression weights that the agents are *implicitly trying to approximate* from their finite in-context samples $(x_i,y_i)$. We, as theorists, use $u^\*$ and $w^\*$ only as reference values so that we can express the fixed points and plateau errors in closed form.
>
> The purpose of the multi-agent analysis is therefore very different: we want to understand *how two agents, each approximating its own objective from its own prompt, interact when these objectives are misaligned*. What the theory reveals is that the final outcome of their interaction is *not* simply a convex combination of $u^\\star$ and $w^\\star$, but depends in a highly structured way on the prompt geometries $S_W, S_U$ and the discrepancy $\\Delta = u^\\star - w^\\star$.
>
> A concrete analogy may help. Suppose in a real negotiation you know that:
> - Agent U wants to *sell* a house for \$800k,
> - Agent W wants to *buy* it for \$600k.
>
> Even if you know the two objectives (the prices), this does *not* tell you where the negotiation will land. The outcome depends on the *process*: how each negotiator speaks, how they respond to each other’s offers, which arguments they emphasize (their geometry), and how aggressively they counter. Two different communication patterns with the *same* underlying objectives can lead to very different agreements.
>
> Our setting is the analogue for LLM agents. The quantities $u^\\star$ and $w^\\star$ let us benchmark what the agents are trying to achieve, but the whole point of our theory is that the *interaction dynamics*, shaped by prompt geometry and misalignment, determine where the agents actually converge. Knowing $u^\\star$ and $w^\\star$ does not tell you where two LLM agents will *end up*; the coupled dynamics do, and this is precisely what Proposition 1 and Corollary 2 characterize.
>
> **Q2 - In Figure 3, it seems that the victim error in LSA-trained agents are high enough, while the attacker’s error is quite low, then, how come the attack success rate is lower than that in GPT5? If this is correct, what are the possible causes? Is it due to differences in model complexity, training data, or other factors? A deeper analysis would enhance the understanding of the results.**
>
> We appreciate the reviewer’s observation. The apparent discrepancy comes from the
> fact that the success-rate metric in the original draft depended on a fixed
> threshold $\\varepsilon$. We realized that the threshold was not matched between the LSA and the GPT-5-mini setting.
>
> Besides, we now believe that reporting a binary success rate was misleading: it obscures the actual phenomenon of interest, which is the structure of the attack trajectories and the induced errors in both the attacker and the victim. In the updated version of the paper, we remove the success-rate diverting the readers from what matters and focus on the continuous error trajectories, which provide a much clearer and model-agnostic comparison between LSA and GPT-5-mini.

---

> ### Author Response · Authors · 2025-11-21
> **Response to reviewer d9dT (Part 3)**
>
> **Q3 - The white-box attack may cause safety issues, are there any solutions or defense approaches? How to secure the multi-agent system or make them more robust? Further, can we derive some insights on what kind of work is suitable for MAS instead of single-agent from this paper?**
>
> *The white-box attack may cause safety issues, are there any solutions or defense approaches?*
>
> In our setting, a white-box attacker designs its geometry $S_U$ using full
> knowledge of $(S_W, w^\\star, u^\\star)$. Algorithm 1 sets
> $$
> S_U \;=\; \\frac{1}{\\eta} P_v + \\varepsilon (I - P_v),
> $$
> $$
> P_v := \\frac{v v^\\top}{\\|v\\|^2},
> $$
> which produces an eigenvalue spike of size $1/\\eta$ on $v$
> and keeps $S_U$ near–isotropic elsewhere. This construction is explicitly
> targeted at the kernel criterion in Proposition 2 / Corollary 5:
> $(I - \\eta S_U) S_W \\Delta \\approx 0$, so that $U$ deletes exactly the
> component $S_W \\Delta$ that $W$ is trying to push, creating the strongest
> possible asymmetry.
>
> The same structure suggests natural defenses. From Proposition~1, both plateau
> errors are quadratic in the misalignment,
> $$\\|w_\\infty - w^\\star\\|_2^2 = \\Delta^\\top S_U S^{-2} S_U \\Delta,$$
>
>
> $$\\|u_\\infty - u^\\star\\|_2^2 \;=\; \\Delta^\\top S_W S^{-2} S_W \\Delta,$$
>
> $$
> S := S_W + S_U,
> $$
> so a first line of defense is to enforce a *small* objective gap
> $\\|\\Delta\\|$ by using a shared global task/safety specification across agents.
> If $\\Delta \\approx 0$, the attack cannot induce a large bias regardless of $S_U$.
>
> Second, the attack crucially relies on placing an eigenvalue essentially at
> $1/\\eta$ along the direction $v = S_W \\Delta$. Indeed,
>
> $$
> r = S_W \\Delta \;-\; \\eta S_U S_W \\Delta,
> $$
>
> so $r$ can be small only if $S_W \\Delta$ is (approximately) an eigenvector of
> $S_U$ with eigenvalue close to $1/\\eta$. In other words, the attacker needs to
> shape its prompt geometry so that $S_U$ acts almost like $(1/\\eta) I$ on the
> particular vector $S_W \\Delta$. If we restrict the attacker’s geometry so that
> its eigenvalues are bounded away from $1/\\eta$ on all directions in the range
> of $S_W$, then $\\|(I - \\eta S_U)S_W \\Delta\\|$ cannot be made
> small. Concretely, the defense mechanism here is to constrain the prompts to avoid extremely
> concentrated examples and regularizing them toward an isotropic form. By Corollary 4, this directly prevents the
> near one-sided convergence exploited by our white-box attack and keeps both
> agents’ plateaus in a controlled regime.
>
> b) *How to secure the multi-agent system or make them more robust?*
>
> We updated the paper to make sure some defense mechanisms for multi-agent systems are discussed:
>
> Beyond the specific white-box construction, our analysis suggests three general
> principles for robust multi-agent design:
>
> (i) *Objective alignment.* Since all plateau errors are quadratic in
> $\\Delta = u^\\star - w^\\star$, the most effective way to reduce vulnerability
> is to enforce a shared global objective across agents (common system-level task
> and safety prompt), so that $\\|\\Delta\\|$ is small. This directly shrinks the
> quadratic forms in Proposition~1 for any attacker geometry.
>
> (ii) *Geometry control.* The strongest instabilities arise when an agent
> can engineer highly anisotropic prompt geometries ($S_W,S_U$) with eigenvalues
> near $1/\\eta$ along misalignment directions. Constraining prompt templates to
> avoid extreme concentration on a single feature direction, and
> regularizing estimated geometries toward more isotropic or bounded-spectral
> forms, keeps $(I - \\eta S_U)S_W \\Delta$ away from zero and thus prevents large asymmetric plateaus.
>
> (iii) *Interaction protocol.* Our dynamics make explicit how alternating
> updates propagate misalignment. In practice, one can reduce this channel by
> limiting how much an untrusted agent can directly overwrite the shared state
> (e.g., mixing updates with a trusted baseline, or routing critical decisions
> through oversight agents whose prompts are deliberately aligned). In all cases,
> the quantities appearing in our theory, $\\Delta$, $S_W$, $S_U$, and the residual
> $r = (I - \\eta S_U)S_W \\Delta$, can be estimated (e.g., via probing) and used as
> diagnostics: if they predict large plateau errors, the configuration is
> structurally fragile and should be revised.
>
> (to be continued)

---

> ### Author Response · Authors · 2025-11-21
> **Response to reviewer d9dT (Part 4)**
>
> (c) *Can we derive some insights on what kind of work is suitable for MAS instead of single-agent from this paper?*
>
> The paper has been revised to provide a clear message on this point. In particular, we now distinguish the fixed-objective and adaptive-objective regimes.
>
> *Fixed-objective:*
>
> If a single agent $W$ runs in isolation on its own quadratic objective, its
> update is
> $$
> w_{t+1} = w_t - \\eta S_W (w_t - w^\\star),
> $$
> which converges to the
> unique minimizer $w^\\star$, so that
>
> $$\\|w_\\infty - w^\\star\\|_2^2 = 0,$$
>
>
> and analogously for $u^\\star$. In the *fixed-objective* multi-agent regime,
> where both agents' targets $(w^\\star,u^\\star)$ are fixed and potentially
> misaligned, Proposition 1 shows that both incur strictly positive plateau
> errors whenever $\\Delta = u^\\star - w^\\star \\neq 0$ and the geometries act
> non-trivially on $\\Delta$:
>
> $$
> \\|w_\\infty - w^\\star\\|_2^2 = \\Delta^\\top S_U S^{-2} S_U \\Delta,
> $$
>
> $$
> \\|u_\\infty - u^\\star\\|_2^2 = \\Delta^\\top S_W S^{-2} S_W \\Delta,
> $$
> $$
> S := S_W + S_U.
> $$
> Thus, in this fixed-target regime, a multi-agent system does *not* improve
> per-agent accuracy on a single underlying objective; it introduces a biased
> compromise between incompatible targets.
>
> *Adaptive-objective:*
>
> By contrast, in the *adaptive* regime **introduced in the revised
> manuscript**, a helper agent is allowed to update its objective turn by turn. The
> cooperative construction in Corollary 3 shows that such a helper can, using only
> turn-local information, provide a Newton-like update for the main agent and
> therefore strictly accelerate convergence (as shown in
> Figure 3). Taken together, this suggests that multi-agent architectures are most suitable either
> for inherently multi-objective or role-structured settings (where a biased
> compromise or division of roles is desired) or for configurations where
> dedicated helper agents can adapt their objectives to speed up or stabilize the
> behavior of a main agent.
>
>
>
>
> We are eager to hear about your further comments. Please let us know if there is anything else we can clarify.

---

> > ### Comment · Reviewer_d9dT · 2025-11-26
> > **Thanks for the response**
> >
> > Thank you for the clarifications regarding Q1, and I think the revision in the paper strengthens its contribution. The added helper-agent corollary and the new experimental results in Figure 3 show that (i) interaction can yield a substantial speed-up even when the true targets are unknown, and (ii) the same analytical framework meaningfully extends to cooperative settings. These additional results address my earlier concerns. I will rise my score.

---

> > > ### Author Response · Authors · 2025-11-28
> > > **Response to reviewer d9dT**
> > >
> > > We are glad that our responses helped clarifying some sections of the paper. We thank you for your constructive and thoughtful suggestions, we appreciate your efforts in helping us improve the quality of our paper.

---

### Official Review · Reviewer_KEzn · 2025-10-31

**Soundness:** 3
**Presentation:** 4
**Contribution:** 3
**Rating:** 6
**Confidence:** 2

**Summary:**

This paper investigates the convergence behavior of two large language model (LLM) agents that perform alternating in-context gradient updates under misaligned objectives, aiming to uncover bias propagation and stability issues in multi-agent interactions. The challenge lies in the strong coupling and low interpretability of multi-agent systems, which make it difficult to mechanistically describe how agents influence each other during in-context updates. The authors model each agent as a linear self-attention (LSA) network that executes gradient descent, formalize their interaction, and derive closed-form and spectral results for the convergence error. They further find that when objective misalignment and prompt-geometry anisotropy coexist, the system exhibits an asymmetric convergence phenomenon. The paper proposes a white-box adversarial prompt design algorithm and validates the “attacker converges, victim fails” behavior on both LSA and GPT-5-mini models. Overall, the paper is well structured with complete and detailed derivations; the direction is promising, though the experimental setup remains somewhat simplified and idealized.

**Strengths:**

1. The paper is the first to model multi-agent LLM interaction as an alternating in-context gradient optimization system, providing a mathematical formulation of inter-agent updates and a computable basis for analyzing bias propagation and convergence stability.
2. The mathematical derivations are complete and logically clear, with consistent notation, explicit assumptions, and boundary conditions; the appendix provides supplementary derivation details that enhance verifiability.
3. The proposed dynamical analysis framework is broadly applicable beyond dual-agent settings—it can extend to multi-agent alignment, model coordination, and optimization-based safety studies, offering a general theoretical tool for future work.

**Weaknesses:**

1. The experiments are conducted only on a synthetic linear-regression task, without testing agent interaction in realistic language scenarios such as reasoning, writing, or code generation. This limits the explanatory power of the results for real-world multi-agent LLM collaboration.
2. The study relies on the assumption that LLM inference is equivalent to in-context gradient descent; while analytically convenient, this assumption is not strictly true for real LLM reasoning and may weaken the practical relevance of the conclusions.
3. Although the paper formally defines prompt geometry, it lacks intuitive examples or visualizations that would help readers understand how the proposed geometric structure corresponds to actual prompt design and model behavior.
4. The writing still needs polishing: the main text frequently refers to GPT-5 while the experiments actually use GPT-5-mini, which should be clarified; moreover, the link in Appendix 7.3 is broken.

**Questions:**

1. Is there sufficient empirical or theoretical evidence for treating LLM inference as in-context gradient descent? Does this assumption still hold on nonlinear tasks?
2. Given the large difference between the LSA model and real LLM architectures, how do the authors evaluate the impact of this simplification on the reliability of their conclusions?
3. Could the authors provide a case study of a large-model experiment?
4. How does “prompt geometry” correspond to linguistic structures? Could the authors give a few simple examples?
5. Could the paper include a more complex experimental scenario, such as realistic multi-agent cooperation (text co-writing, code generation, etc.)?
6. Do the authors plan to release the LSA experiment code or a minimal reproducible example?
7. If the attacker only has black-box access to the victim, is there an approximate attack strategy?

---

> ### Author Response · Authors · 2025-11-21
> **Response to reviewer KEzn (Part 1)**
>
> We sincerely thank the reviewer for the insightful and constructive feedback. We appreciate your careful reading of our work and your helpful suggestions for improving the paper.
>
> **W1 - The experiments are conducted only on a synthetic linear-regression task, without testing agent interaction in realistic language scenarios such as reasoning, writing, or code generation. This limits the explanatory power of the results for real-world multi-agent LLM collaboration.**
> We agree that our experiments are restricted to synthetic in-context linear regression, and we appreciate the opportunity to clarify the intended scope. Our goal in this paper is *not* to provide a broad empirical evaluation of multi-agent LLM systems across realistic tasks; that is precisely what several recent works on multi-agent LLM collaboration already focus on (e.g., debate-style setups, role-structured teams, and code- or reasoning-oriented agent collectives). Instead, our contribution is complementary and deliberately focuses on theory, which is a missing piece for multi-agent systems. Specifically, to the best of our knowledge, there is no theoretical formalism that enables one to study them from first principles, and this is what we propose with our paper.
> Specifically, we adopt an *in-context optimization* perspective and instantiate it in a linear self-attention (LSA) model of two interacting in-context gradient-based learners on a quadratic regression task. This setting is intentionally simplified and aligned with the in-context learning literature: it matches the assumptions of our theoretical analysis and allows us to derive closed-form expressions for the coupled dynamics and plateau biases, which are currently out of reach for full-scale LLMs on open-ended language tasks. The synthetic experiments (including the GPT-5 mini runs on the same linear-regression tasks) are therefore meant as a controlled testbed for validating the predictions of this LSA in-context optimization framework, rather than as realistic multi-agent benchmarks.
>
> **W2 - The study relies on the assumption that LLM inference is equivalent to in-context gradient descent; while analytically convenient, this assumption is not strictly true for real LLM reasoning and may weaken the practical relevance of the conclusions.**
> A growing body of recent work shows that transformers with chain-of-thought or algorithmic prompting can implement a broad class of algorithms in context, not just linear regression. For example, “Transformers Learn In-Context by Gradient Descent” (von Oswald et al., 2023) and “One Step of Gradient Descent is Provably the Optimal In-Context Learner with One Layer of Linear Self-Attention” (Mahankali et al., 2023) prove that linear self-attention architectures can exactly implement gradient-descent updates from in-context data. “What Learning Algorithm Is In-Context Learning? Investigations with Linear Models” (Akyürek et al., 2023) shows that trained transformers recover gradient descent (or closely related algorithms) when solving linear tasks from examples, while “Linear Transformers Are Versatile In-Context Learners” (Ahn et al., 2023) demonstrates that linear transformer variants learn a range of optimization-like procedures in context. More broadly, “Transformers as Algorithms: Generalization and Stability in In-Context Learning” (Li et al., 2024) studies transformers that implement algorithmic procedures (including optimization) in context and analyzes when such behaviors generalize.
> Within this landscape, our choice to focus on gradient descent for linear regression should be read as selecting a well-understood mechanistic testbed, rather than postulating an arbitrary update rule. The specific setting, single-layer linear self-attention implementing gradient descent on a quadratic objective lets us write down closed-form dynamics for two interacting in-context optimizers, which is currently out of reach for full-scale LLMs on open-ended language tasks.
> That said, we agree that treating LLM inference as literally equivalent to in-context gradient descent everywhere would be too strong. Our intention is more modest: we adopt in-context gradient descent as a modeling idealization that is (i) directly supported on linear and related tasks by the works cited above, and (ii) analytically tractable enough to expose the geometric structure of multi-agent interaction. We have revised the manuscript to make this more explicit.

---

> ### Author Response · Authors · 2025-11-21
> **Response to reviewer KEzn (Part 2)**
>
> **W3 - Although the paper formally defines prompt geometry, it lacks intuitive examples or visualizations that would help readers understand how the proposed geometric structure corresponds to actual prompt design and model behavior.**
>
> Conceptually, the notion is very direct: for an LLM, a fixed prompt induces a collection of embeddings, and the prompt geometry is simply the covariance structure of these representations (which directions are most activated, how anisotropic that distribution is, etc.). Our LSA model is an idealized version of this: instead of passing tokens through an embedding mapping, we work directly with feature vectors. We clarify this in the paper, and answer more thoroughly regarding this point in the question answers.
>
>
> **Q1 - Is there sufficient empirical or theoretical evidence for treating LLM inference as in-context gradient descent? Does this assumption still hold on nonlinear tasks?**
>
> There is now a substantial body of work, both constructive and empirical, supporting the use of in-context gradient descent as a mechanistic model for how transformers can perform in-context learning on supervised tasks. For example, Transformers Learn In-Context by Gradient Descent” (von Oswald et al., 2023), give explicit weight constructions showing that a linear self-attention transformer can implement gradient descent on regression losses, and show that trained transformers closely match GD iterates on linear and nonlinear regression via deep representations. “Transformers as Statisticians: Provable In-Context Learning with In-Context Algorithm Selection” (Bai et al. 2023) prove that transformers can implement a broad class of standard algorithms in context, including least squares, generalized linear models, and gradient descent on two-layer neural networks. “Transformers learn to implement preconditioned gradient descent for in-context learning” (Ahn et al. 2023) show that linear transformers trained on random regression problems learn to implement preconditioned gradient descent. Empirically, “Trained Transformers Learn Linear Models In-Context” (Zhang et al. 2024) and related work further find that trained transformers learn linear models in context with behavior that matches gradient descent baselines.
> These results extend beyond purely linear tasks: the constructions in Bai et al. (2023) and von Oswald et al. (2023) cover gradient descent on certain nonlinear models (e.g., two-layer networks, linear models on deep representations), and more recent work shows transformers can even implement multi-step optimization procedures in context (Huang et al.)
>
> Our work leverages this recent body of work to provide a novel point of view on multi-agent framework that can be analytically explained and explored. In fact, our assumption should be read as a modeling idealization rather than a literal claim about all LLM behavior. We focus on a regime where treating each agent as an in-context gradient-based optimizer is well-motivated by prior work, and we make this assumption explicit by working in an LSA model that exactly instantiates in-context GD. Our conclusions are therefore conditional: when an LLM agent’s forward pass can be well-approximated by gradient-like updates under some effective geometry, our analysis predicts how objective misalignment and prompt geometry shape the resulting multi-agent dynamics.
>
> We updated the paper to make sure this message is clear (abstract, intro, conclusion, limitation and scope section)

---

> ### Author Response · Authors · 2025-11-21
> **Response to reviewer KEzn (Part 3)**
>
> **Q2 - Given the large difference between the LSA model and real LLM architectures, how do the authors evaluate the impact of this simplification on the reliability of their conclusions?**
>
> We agree that there is a substantial gap between LSA models and full-scale LLM architectures, and we do not claim that the LSA setting quantitatively captures all aspects of real LLM behavior. Our goal is instead to isolate and analyze a small number of structural ingredients that are shared with more realistic systems: (i) an in-context optimization perspective (agents behave approximately like gradient-based learners on an implicit objective), (ii) anisotropic prompt geometry that weights different directions in representation space unevenly, and (iii) alternating updates between agents with potentially misaligned objectives. The LSA model is chosen because it makes these ingredients mathematically tractable, allowing us to derive closed-form fixed points and plateau errors. The connection between our LSA conclusions and real LLMs have been made more explicit in the revised paper (mainly in Section 3).
>
>
> **Q3 -Could the authors provide a case study of a large-model experiment?**
>
> We agree that large-model case studies are valuable for grounding mechanistic analyses, and we have taken a first step in this direction with our GPT-5-mini experiments on the same in-context linear-regression tasks used in our theory. There, we treat GPT-5-mini as a black-box agent and vary the degree of objective misalignment and prompt geometry, observing qualitatively consistent patterns with our LSA predictions. Due to computational and space constraints, we did not include a richer, task-specific case study in this work. Also we explicitly highlight that extending the analysis to more realistic multi-agent LLM tasks as an important direction for future work. Finally, we want to reinforce the fact that this work purpose is to give a theoretical test bed to the analysis of multi-agent following the in-context learning literature and not to provide empirical evaluations of multi-agent systems.
>
>
> **Q4 - How does “prompt geometry” correspond to linguistic structures? Could the authors give a few simple examples?**
>
> In real LLMs, a prompt is first mapped to a sequence of hidden representations (embeddings) $h_1, \\dots, h_n \\in \\mathbb{R}^d$ at some layer. The prompt geometry is the second-moment structure of these vectors, e.g. $S =  \\frac{1}{n} \\sum_i h_i h_i^\\top$. Linguistic structures appear as dominant directions and anisotropies in this covariance: directions in representation space that are repeatedly activated by particular lexical, syntactic, or semantic patterns in the prompt.
>
> Concretely, our LSA matrices $S_W, S_U$ play the role of such covariances in an idealized setting where the “hidden states” are directly given as feature vectors. For intuition, consider three simple examples in a real LLM:
>
> A prompt consisting mostly of numerical arithmetic instructions and examples (“add”, “sum”, short equations) will produce hidden states concentrated along directions encoding numerical/arithmetical structure, making $S$ highly anisotropic along an “arithmetic” subspace.
>
> A prompt filled with Python code snippets and API names will yield hidden states clustered along directions representing programming syntax and identifiers; the corresponding $S$ emphasizes those “code” directions and de-emphasizes everyday language.
>
> A prompt dominated by legalistic or formal phrases (“hereby”, “pursuant to”, “the party of the first part…”) will produce hidden states aligned with a legal style subspace, whereas a casual conversational prompt will emphasize a different region of representation space.
>
> In all cases, the prompt geometry captures which linguistic structures (semantic fields, styles, task types) are instantiated in the prompt. In the revised manuscript, we clarify this by explicitly describing $S_W, S_U$ as covariances of prompt-induced representations and adding a short paragraph connecting these covariances to such simple linguistic examples.
>
> We updated section 3 to clarify this.

---

> ### Author Response · Authors · 2025-11-21
> **Response to reviewer KEzn (Part 4)**
>
> **Q5 - Could the paper include a more complex experimental scenario, such as realistic multi-agent cooperation (text co-writing, code generation, etc.)?**
>
> We agree that realistic multi-agent cooperation tasks (text co-writing, code generation, etc.) are important for evaluating practical systems and enriching the theoretical testbed we propose. However, they are intentionally not the focus of this work. Our goal here is theory first and extending the in-context learning body of work: we study a setting where we can model each agent as an in-context optimizer in an LSA framework. Introducing rich, open-ended multi-agent benchmarks would require additional design choices (task selection, evaluation metrics, prompt engineering) that are pretty far from our main contribution and would substantially expand the scope of the paper. We instead view our analysis as providing mechanistic, testable predictions that future work can probe in realistic multi-agent LLM cooperation settings. Also, from a pragmatic standpoint, if we were doing this real scale experiment, because of the size limit, we would need to remove at least an entire section of the current paper, which will make the paper logic really hard to follow. However, we agree that  this is clearly a future work direction that we plan to investigate.
>
>
> **Q6 - Do the authors plan to release the LSA experiment code or a minimal reproducible example?**
>
> The code was submitted together with the paper and is provided in the supplementary-material. It is fully accessible to the reviewers and will become publicly available at the end of the rebuttal phase.
>
>
> **Q7 - If the attacker only has black-box access to the victim, is there an approximate attack strategy?**
>
> Our formal attack algorithm (Algorithm 1) is indeed derived in a white-box
> setting, where the attacker knows $S_W$, and objective spread $\\Delta$. In practice, the attacker does not need literal access to the victim’s parameters. What matters is an approximation to the victim’s objective direction and prompt geometry. We see two plausible black-box routes:
>
> 1. Prior knowledge: In many realistic applications, the victim’s objective and prompt are highly constrained and largely guessable (e.g., a hotel booking agent wants to charge the highest price possible, a customer agent wants to buy goods for the cheapest price, … ). This already gives the attacker a strong prior on the form of the victim’s objective direction and typical prompt geometry (what features / semantic directions are emphasized), allowing them to build the approximates $\\hat{S}_W$ and $\\hat{\\Delta}$, which can then be plugged as is into the white-box attack algorithm we propose.
>
> 2. Probe estimation: Even without such prior knowledge, a black-box attacker can interact with the victim and basically perform prompt hacking as to estimate the objective direction and prompt geometry. Again, these estimates $\\hat{S}_W$ and $\\hat{\\Delta}$) can be directly used to construct the attack algorithm we provide.
>
> We are eager to hear about your further comments. Please let us know if there is anything else we can clarify.

---

> ### Comment · Reviewer_KEzn · 2025-11-28
>
> Thank you for your responses. Some of my concerns have been resolved, and I will consider raising my rating.

---

> > ### Author Response · Authors · 2025-11-28
> > **Response to reviewer KEzn**
> >
> > Thank you for your constructive and thoughtful suggestions, we appreciate your efforts in helping us improve the quality of our paper.

---

### Official Review · Reviewer_HbCZ · 2025-11-02

**Soundness:** 3
**Presentation:** 3
**Contribution:** 2
**Rating:** 6
**Confidence:** 3

**Summary:**

The paper studies the dynamics of two single-layer transformers with linear self-attention (LSA agents) alternately performing in-context gradient descent toward their own objectives. Theoretical analysis show they fall in different regime depending on different configurations of the misaligned objectives. Experiments on two LLMs doing in-context gradient descent validates the theoretical results.

**Strengths:**

This paper shows an interesting angle to study agent-to-agent interactions through in-context gradient descents of LSAs. The theoretical results show that misaligned objective correspond to different behavior. The experiment also generalizes it to LLMs (GPT5) that validates the theoretical analysis. The paper is well-written.

**Weaknesses:**

* Investigating multi-LLM-agents interactions is an important and emerging problem. Although this paper offers an interesting perspective, it builds on oversimplified settings that is not obvious to generalize easily. I appreciate the authors ackoknledging this in the conclusion: "move beyond controlled linear tasks and examine these mechanisms directly in large-scale LLMs." However, I believe this is should be an important point and worth discussing in more detail.
* There is some degree of over-claiming: the abstract reads like the theory is developed for generic LLMs, while the theory is actually developed for LSAs.

**Questions:**

* Line 276: "These results suggest a concrete prompt-design principle for multi-agent systems..." Can the authors be more concrete about what specific results suggest these concrete prompt-design principles, and how?
* This paper suggests (e.g., from Proposition 1 or Figure 1) that the multi-agent system has non-zero error w.r.t. to each agent's respective objective. This seems to imply that there is not benefit of using a multi-agent system. In the LSA in-context gradient descent setting, are there circumstances where the multi-agent system can be more beneficial than using each agent separately?

---

> ### Author Response · Authors · 2025-11-21
> **Response to reviewer HbCZ (Part 1)**
>
> We sincerely thank the reviewer for the insightful and constructive feedback. We appreciate your careful reading of our work and your helpful suggestions for improving the paper. We agree that our setting is intentionally simplified, and we have revised the manuscript to make this scope explicit and to better articulate how the framework connects to real LLM behavior.
>
>
> **W1 - Investigating multi-LLM-agents interactions is an important and emerging problem. Although this paper offers an interesting perspective, it builds on oversimplified settings that is not obvious to generalize easily. I appreciate the authors ackoknledging this in the conclusion: "move beyond controlled linear tasks and examine these mechanisms directly in large-scale LLMs." However, I believe this is should be an important point and worth discussing in more detail.**
>
>
>
> First, we added a dedicated explanation in Section~3 describing how *prompt geometry* manifests in actual LLMs. We now make explicit that the LSA matrices $S_W, S_U$ correspond to the second-moment structure of LLM embeddings, and we provide concrete examples (arithmetic prompts, code prompts, legalistic text) illustrating how different linguistic structures induce different anisotropic geometries in representation space. This addition directly clarifies how the simplified model relates to real LLM prompt design.
>
> Second, we strengthened the conceptual bridge between the simplified setting and real multi-agent systems by introducing an explicit distinction between two regimes:
> (i) the *fixed-objective* regime, where misalignment necessarily produces biased equilibria, and
> (ii) the newly added *adaptive-objective* regime, where a helper agent can dynamically shape a turn-local objective and implement a Newton-like update using only observable interaction state $(u_t, w_{t+1}, S_W, S_U, \eta)$. This clarifies when multi-agent systems can worsen or improve performance, and why similar phenomena can appear in more realistic LLM settings where agents can be instructed to play helper roles.
>
> **W2 - There is some degree of over-claiming: the abstract reads like the theory is developed for generic LLMs, while the theory is actually developed for LSAs.**
>
> Here are the main modifications to the papers that should help improve the paper on this point:
>
> a) we update the abstract:
>
> We develop and analyze a theoretical framework for agent-to-agent interactions in a simplified in-context linear regression setting. In our model, each agent is instantiated as a single-layer transformer with linear self-attention (LSA) trained to implement gradient-descent-like updates on a quadratic regression objective from in-context examples. We then study the coupled dynamics when two such LSA agents alternately update from each other’s outputs under potentially misaligned fixed objectives. Within this framework, we characterize the generation dynamics and show that misalignment leads to a biased equilibrium where neither agent reaches its target, with residual errors predictable from the objective gap and the prompt-induced geometry. We further contrast this fixed objective regime with an adaptive multi-agent setting, wherein a helper agent updates a turn-based objective to implement a
> Newton-like step for the main agent, eliminating the plateau and accelerating its convergence. Experiments with trained LSA agents, as well as black-box GPT-5-mini runs on in-context linear regression tasks, are consistent with our theoretical predictions within this simplified setting. We view our framework as a mechanistic framework that links prompt geometry and objective misalignment to stability, bias, and robustness, and as a stepping stone toward analyzing more realistic multi-agent LLM systems.
>
>
> b) Within the introduction, when we introduce the in-context learning literature framework we are opting for, we added:
>
> In the rest of the paper, “agent” refers specifically to such an LSA-based in-context optimizer operating on a linear regression objective. We use this analytically tractable model as a proxy for LLM-based agents.

---

> ### Author Response · Authors · 2025-11-21
> **Response to reviewer HbCZ (Part 2)**
>
> **Q1 - Line 276: "These results suggest a concrete prompt-design principle for multi-agent systems..." Can the authors be more concrete about what specific results suggest these concrete prompt-design principles, and how?**
>
> We agree that the original phrasing did not clearly indicate which results support the prompt-design principle or how they
> do so. In the revised version, we substantially clarified this point in two
> ways and added new paragraphs in Section 3 to make the connection explicit:
>
> "
> From Proposition 1, the asymptotic squared errors can be written as
>
> $$\\|u_\\infty - u^\\star\\|_2^2 = \\Delta^\\top S_W S^{-2} S_W \\Delta,$$
>
> $$\\|w_\\infty - w^\\star\\|_2^2 = \\Delta^\\top S_U S^{-2} S_U \\Delta,$$
>
> where $\\Delta = u^\\star - w^\\star$ encodes the discrepancy between the two
> prompt-induced objectives and $S_W, S_U$ are determined by the prompt geometries.
> Thus, for fixed prompt geometries, both agents’ plateau errors grow quadratically with the size
> of the objective gap $\\|\\Delta\\|_2$, and are further amplified when $\\Delta$ has large components
> along eigen-directions where $S_W S^{-2} S_W$ or $S_U S^{-2} S_U$ have large eigenvalues.
> Corollary 2 then shows that, after normalizing by $\\|w^\\star\\|_2^2 + \\|u^\\star\\|_2^2$, these
> plateau errors are nondecreasing functions of the alignment angle $\\theta$ between $w^\\star$ and $u^\\star$.
> In our LSA model, both $\\Delta$ and the geometries $S_W, S_U$ are fully determined by the
> prompts (system instructions and in-context examples) given to each LSA agent.
>
>
> These results yield a concrete prompt-design principle for multi-agent systems: construct the
> system and task prompts so that the effective objectives realized in-context are as aligned as
> possible (small $\\|\\Delta\\|_2$ and small $\\theta$), for example by explicitly encoding a shared
> global objective and avoiding components that pull $w^\\star$ and $u^\\star$
> in different directions.
> “
>
>
> We also added the following to help the reader understand the bridge between LSA and LLM:
>
>
> “In an LLM, an analogous notion of prompt geometry can be defined directly
> in representation space. Given a prompt $P = (t_1,\\dots,t_n)$ and its embedding
> representations $h_1,\\dots,h_n \\in \\mathbb{R}^d$, one can
> form a second-moment matrix $S(P) = \\frac{1}{n} \\sum_{i=1}^n h_i h_i^\\top$, whose dominant directions and anisotropy reflect which linguistic structures
> (semantic fields, styles, task types) are repeatedly instantiated by the
> prompt. For example, a prompt dominated by arithmetic expressions, by code
> snippets, or by legalistic text will emphasize different subspaces in
> representation space. In our LSA model, the matrices $S_W$ and $S_U$ play this
> role for the feature vectors appearing in each agent’s prompt.
> “

---

> ### Author Response · Authors · 2025-11-21
> **Response to reviewer HbCZ (Part 3)**
>
> **Q2 - This paper suggests (e.g., from Proposition 1 or Figure 1) that the multi-agent system has non-zero error w.r.t. to each agent's respective objective. This seems to imply that there is not benefit of using a multi-agent system. In the LSA in-context gradient descent setting, are there circumstances where the multi-agent system can be more beneficial than using each agent separately?**
>
>
> This is a great point that we now clarified in the updated version. In our LSA in-context gradient descent model, if one measures performances
> by each agent’s squared error to its *own fixed global* quadratic objective, then
> the reviewer’s interpretation is correct in the *fixed-objective* regime:
> the multi-agent dynamics do not yield a per-agent performance improvement over
> running each agent in isolation.
>
> Concretely, when an agent runs alone on its own data, the update reduces to
> standard gradient descent on its least-squares loss. For example, for agent $W$
> we recover
>
> $$
> w_{t+1} = w_t - \\eta S_W (w_t - w^\\star),
> $$
>
> which converges to the unique minimizer $w^\\star$, so the asymptotic error of
> the single-agent baseline is $\\|w_\\infty - w^\\star\\|_2^2 = 0$ (and analogously
> for agent $U$). In contrast, Proposition 1 shows that under agent-to-agent
> interaction with misaligned fixed objectives $\\Delta = u^\\star - w^\\star \\neq 0$,
>
> $$ \\|u_\\infty - u^\\star\\|_2^2  = \\Delta^\\top S_W S^{-2} S_W \\Delta + O(\\eta)$$
>
> $$\\|w_\\infty - w^\\star\\|_2^2 = \\Delta^\\top S_U S^{-2} S_U \\Delta + O(\\eta)$$
>
> As a result, whenever $\\Delta$ has non-zero component along
> directions where the prompt covariances act, both agents necessarily incur a
> strictly positive plateau error, which is shown in Figure 1.
>
> Besides, in the revised version we now explicitly distinguish this fixed-objective
> regime from an *adaptive* regime in which one agent can update its
> objective turn by turn. In the latter regime, Corollary 3 provides a concrete cooperative construction: a helper agent $U$ uses only turn-based quantities $(u_t,w_{t+1},S_W,S_U,\\eta)$
> to choose a dynamic target $u_t^\\star$ such that its realized update implements
> a Newton step for $W$, driving the system directly to $w^\\star$. Our new
> experiment in Figure 3 shows that a small number of such helper-assisted steps can match or outperform hundreds of single-agent  steps. Thus, while misaligned fixed objectives
> necessarily hurt per-agent performance, allowing a cooperative helper to adapt
> its objective can strictly *improve* convergence rate compared to a single agent.
>
> From a practical perspective, this dichotomy mirrors real multi-agent systems:
> when agents are locked into conflicting goals, interaction produces biased
> compromises. Now, when some agents are explicitly tasked with helping others (e.g.,
> critique, verification, or correction roles), and can *adapt* their intermediate
> objectives based on turn-local information, multi-agent interaction can
> drastically improve reliability and efficiency.
>
>
> We are eager to hear about your further comments. Please let us know if there is anything else we can clarify.

---

### Author Response · Authors · 2025-11-21
**Global Response (by authors)**

We sincerely thank all the reviewers for their insightful and constructive comments. Following your suggestions, we have revised the manuscript to clarify the scope of our framework, tighten our claims, and better connect the theory, experiments, and practical implications.

In this update, we made the following main changes:

- **Clarified scope, modeling assumptions, and limitations.**
  We rewrote the abstract and the opening of Section 1 to state explicitly that our theory is developed for a simplified in-context *linear regression* setting with single-layer linear self-attention (LSA) agents, used as a mechanistic proxy for LLM-based agents rather than as a model of arbitrary LLM behavior. We also added a dedicated “Limitations and scope” paragraph before the conclusion, emphasizing that our goal is to provide a controlled theoretical testbed that complements existing empirical work on realistic multi-agent LLM systems, not to claim direct quantitative predictions for all such systems.

- **Made the prompt-design principles and multi-agent benefits explicit.**
  In the fixed-objective regime, we now clearly tie the prompt-design principle to Proposition 1 and Corollary 2, which show how plateau errors depend quadratically on the objective gap $\Delta$ and on the prompt geometries $S_W, S_U$. In addition, we introduced a new subsection on *adaptive helper objectives* and a new result (Corollary 3), showing how a helper agent can use turn-local information to implement a Newton-like step for the main agent. The new experiment in Figure 3 illustrates that this adaptive regime turns the same agent-to-agent interface from mutual degradation into cooperative acceleration.

- **Strengthened the discussion of the in-context GD assumption.**
  We expanded the related-work and limitations discussion to situate our “transformers-as-optimizers” perspective within recent in-context learning theory. We now emphasize that treating agents as in-context gradient-based optimizers is a *modeling idealization* supported on linear and related tasks by prior work, and that our conclusions should be read as conditional on this regime rather than as claims about arbitrary LLM reasoning.

- **Clarified prompt geometry and polished experiments.**
  We added an intuitive representation-space interpretation of prompt geometry $S_W, S_U$ (via second moments of prompt-induced embeddings) and connected it to concrete linguistic examples. In the experimental section, we removed the potentially misleading binary “success-rate” metric for attacks and now focus on continuous error trajectories for both LSA agents and GPT-5-mini, which more directly reflect the dynamics predicted by our theory.

We believe these revisions substantially improve clarity, scope, and practical relevance and strengthen the state of our submission. We thank the reviewers again for their detailed feedback and welcome any further suggestions.

We also want to re-state our contribution. Note that our work provides a first mechanistic bridge between prompt geometry, objective misalignment, and multi-agent stability, a direction that is currently, to the best of our knowledge, missing in the literature and complementary to existing empirical evaluations of multi-agent LLMs.

---

## Summary of Contributions

- **A tractable dynamical model of multi-agent interaction.**
  We instantiate each agent as a linear self-attention (LSA) in-context optimizer performing gradient-descent updates on a quadratic regression task, giving a closed-form, analyzable model of agent-to-agent update dynamics.

- **A complete characterization of fixed-objective interaction.**
  We derive exact expressions (Proposition 1, Corollary 2) for the biased equilibrium reached when two agents with misaligned objectives alternately update each other, showing how plateau errors depend quadratically on objective misalignment and on prompt-induced geometry.

- **A new cooperative regime with adaptive helper objectives.**
  We introduce an *adaptive* variant of multi-agent interaction and prove (Corollary 3) that a helper agent can use turn-based information to implement a Newton-like update for the main agent, eliminating the bias and accelerating convergence. This identifies a concrete mechanism through which multi-agent systems can outperform single agents.

- **A theoretically grounded adversarial construction.**
  We derive the unique geometry-based condition enabling asymmetric convergence (Proposition 2) and demonstrate white-box attacks that exploit misalignment.

- **Empirical validation in a controlled testbed.**
  We validate the theoretical predictions using trained LSA agents and black-box GPT-5-mini on in-context linear regression tasks, showing that the predicted adversarial behaviors manifest even in large models under controlled conditions.

---

### Author Response · Authors · 2025-12-01
**Summary of rebuttal period (part 3)**

### 3) Collaborative agent regime and implications for practical MAS

A separate set of concerns (especially from HbCZ, d9dT, dPdy) asked whether our analysis only shows that multi-agent systems harm performance, and how our theory can inform *collaborative* multi-agent design and failure mitigation (including recent work like Aegis and MAST).

To address this, we **pulled out and strengthened the collaborative “helper agent” story** as its own regime:

- We introduced a **new adaptive multi-agent regime** with a **helper agent** and a new theoretical result (**Corollary 3**):
  - In contrast to the fixed-objective regime (where misalignment necessarily produces biased equilibria), the helper agent is allowed to adapt a **turn-based objective** using only observable interaction state.
  - Corollary 3 shows that the helper can choose its turn-local target so that its realized update implements a **Newton-like step** for the main agent—eliminating the plateau and significantly **accelerating convergence**.

- We added a **new experiment (Figure 3)** demonstrating that:
  - A small number of helper-assisted steps can match or outperform hundreds of single-agent steps.
  - This directly answers the question of when multi-agent systems can **outperform** a single agent in our framework.

- We explicitly connect this adaptive regime to **practical MAS design**:
  - We relate our fixed-objective misalignment analysis to **MAST**’s findings about system-design and inter-agent misalignment failures, interpreting misaligned prompts and roles as structural sources of biased equilibria.
  - We interpret the helper regime as a concrete mechanism for repairing such failures: the helper’s objective is explicitly conditioned on the main agent’s state, ensuring “messages” are exactly the information the main agent needs, aligning with MAST’s call for better role design and information flow.
  - We also connect to **Aegis** by interpreting perturbations to intermediate messages as perturbations to the shared state in our dynamics, and explain how misalignment and anisotropic geometries amplify such perturbations over multiple turns.

This collaborative agent regime makes clear that our framework is not only about failure modes and attacks: it also provides **constructive guidance** on when and how multi-agent architectures (with explicit helper roles) can be structurally beneficial.

---

**Reviewer updates**

After these revisions and clarifications:

- **Reviewer d9dT** explicitly states that the new helper-agent corollary and experiments strengthen the contribution, that earlier concerns are resolved, and they have **raised their score**.
- **Reviewer dPdy** reports that their main concerns (about the GPT-5 setup, robustness, and practical connections) have been addressed, appreciates the strengthened link to MAS failure-attribution work, and has **raised their score**.
- **Reviewer KEzn** indicates that several concerns have been resolved and notes they **will consider raising their rating**; their latest comment is positive about the revisions and our clarifications.
- **Reviewer HbCZ** already had a positive score, and their main worries about scope have been directly addressed by the revised abstract, introduction, and limitations/scope discussion.

In summary, the revised paper offers a clearer, more precisely scoped, and technically stronger theoretical framework for mechanistic analysis of multi-agent interactions, with additional theory (Corollaries 3 and 4), new supporting experiments, and an explicit bridge to practical MAS design and safety. Two reviewers have already raised their scores and a third has signaled an intention to do so.

---

### Author Response · Authors · 2025-12-01
**Summary of rebuttal period (part 2)**

### 2) Robustness, attacks vs defenses, and experimental metrics

Reviewers (particularly d9dT and dPdy) also pressed on how robust asymmetric convergence is beyond the exact algebraic conditions, how to interpret our empirical attack results, and what defenses our analysis suggests.

- We added **Corollary 4** to analyze **approximate asymmetric convergence**:
  - It quantifies how the plateaus change when the kernel condition underlying Proposition 2 is only approximately met, showing that:
    - The attacker’s plateau remains close to its exact-case value up to a controlled correction;
    - The victim’s plateau grows smoothly with the residual in the alignment condition.
  - This directly addresses concerns about brittleness: asymmetric convergence degrades **smoothly** rather than disappearing abruptly.

- We expanded the **defense and robustness discussion**, extracting concrete design principles:
  - From Proposition 1, plateau errors are quadratic in the objective gap, which suggests enforcing a **shared global objective** (or strong safety/task alignment) across agents to keep the gap small.
  - From Proposition 2 and the eigen-spike construction, we identify that the attack depends on placing eigenvalues near \(1/\eta\) along specific misalignment directions; thus, constraining prompt geometries (e.g., avoiding extremely concentrated, highly anisotropic prompts and regularizing toward bounded spectral norms) help mitigates the attack.
  - We also discuss **interaction protocol defenses**, such as limiting how much untrusted agents can overwrite shared state and routing critical decisions through more trusted/aligned agents.

- We clarified the **intended role of the GPT-5-mini experiment**:
  - It is not meant to re-establish emergent in-context optimization (which is already covered in prior work), but to test whether, *given* a gradient-like agent, adversarial prompt geometries still induce the asymmetric dynamics predicted by our theory in a realistic black-box model.
  - We emphasize that a large LLM *could* in principle damp or regularize such attacks, and that the fact GPT-5-mini does not fully do so highlights a concrete potential risk for multi-agent LLM systems.

- We **removed the binary “attack success-rate” metric**:
  - We realized that this metric (and mismatched thresholds between LSA and GPT-5-mini) was misleading and hard to interpret.
  - The revised paper now focuses on **continuous error trajectories** for both LSA agents and GPT-5-mini, which are reproducible and directly aligned with the theoretical predictions.

---

---

### Author Response · Authors · 2025-12-01
**Summary of rebuttal (part 1)**

**Summary of concerns, revisions, and reviewer updates**

We would like to summarize  the rebuttal period as to best help the AC to understand how we updated our work.

Reviewers raised concerns around (i) how our prompt-geometry framework and assumptions relate to real multi-agent LLM systems, (ii) the robustness of our asymmetric-convergence / white-box attack analysis and how to interpret the experiments, and (iii) how our theory connects to *collaborative* multi-agent design beyond adversarial settings. We address these below.

---

### 1) Prompt geometry, benefits of multi-agent systems, and new theory/experiments

Several reviewers (HbCZ, KEzn, d9dT, dPdy) asked for clearer intuition on prompt geometry, a more precise statement of scope (LSA vs generic LLMs), and a sharper explanation of when multi-agent interaction helps or hurts.

- We added an **intuitive representation-space interpretation of prompt geometry**:
  - Prompt geometry is now described as the **second-moment structure of prompt-induced embeddings**, with concrete examples (arithmetic-heavy prompts, code prompts, legalistic text) that illustrate how different linguistic structures induce different anisotropic geometries.
  - We explicitly connect these ideas to the LSA matrices in our model, which play the role of these covariance structures.

- We made the **prompt-design principle** precise and explicitly tied it to **Proposition 1 and Corollary 2**:
  - Proposition 1 shows that both agents’ plateau errors grow quadratically with the **objective gap** and are further amplified when the misalignment vector has large components along eigen-directions of the prompt geometries.
  - Corollary 2 shows that, after normalizing by the gap, these plateau errors are nondecreasing in an **alignment angle** between the effective objectives.
  - We now spell out the resulting **prompt-design guideline**: construct system/task prompts so that the in-context effective objectives are as aligned as possible (small gap and small angle), e.g., via explicit shared global objectives and by avoiding instructions that pull the agents in systematically different directions.

- We **clarified scope** by rewriting the **abstract** and the opening of **Section 1** to explicitly state that:
  - Our theory is developed in a **simplified in-context linear regression setting** with **single-layer LSA agents**;
  - These agents are used as a **mechanistic proxy** for LLM-based agents, *not* as a quantitative model of arbitrary LLM behavior;
  - We added an explicit **“limitations and scope”** paragraph that positions the work as a controlled theoretical testbed complementary to empirical multi-agent LLM studies.

Overall, this set of changes directly addresses concerns about interpretability of prompt geometry, scope, and the conditions under which multi-agent interaction is helpful vs. harmful in our framework.

---

---

### Note · Program_Chairs · 2026-01-17
**Submission Desk Rejected by Program Chairs**

The following references in this submission do not refer to real documents and/or have major errors in bibliographic information:

 Andy Zou, Zifan Chen, Alexander Yang, Eric Zhang, Nicholas Carlini, Daphne Ippolito, John Lee, Xiang Lisa Li, Michael Zhang, Matt Fredrikson, et al. Effective prompt extraction from language models. In Proceedings of the 62nd Annual Meeting of the Association for Computational Linguistics (ACL), 2023.